# The Swr1 chromatin-remodeling complex prevents genome instability induced by replication fork progression defects

Anjana Srivatsan [1], Bin-Zhong Li[1], Barnabas Szakal [2], Dana Branzei [2,3], Christopher D. Putnam [1,4] & Richard D. Kolodner [1,5,6,7]

Genome instability is associated with tumorigenesis. Here, we identify a role for the histone Htz1, which is deposited by the Swr1 chromatin-remodeling complex (SWR-C), in preventing genome instability in the absence of the replication fork/replication checkpoint proteins Mrc1, Csm3, or Tof1. When combined with deletion of *SWR1* or *HTZ1*, deletion of *MRC1*, *CSM3*, or *TOF1* or a replication-defective *mrc1* mutation causes synergistic increases in gross chromosomal rearrangement (GCR) rates, accumulation of a broad spectrum of GCRs, and hypersensitivity to replication stress. The double mutants have severe replication defects and accumulate aberrant replication intermediates. None of the individual mutations cause large increases in GCR rates; however, defects in *MRC1*, *CSM3* or *TOF1* cause activation of the DNA damage checkpoint and replication defects. We propose a model in which Htz1 deposition and retention in chromatin prevents transiently stalled replication forks that occur in *mrc1*, *tof1*, or *csm3* mutants from being converted to DNA double-strand breaks that trigger genome instability.

[1] Ludwig Institute for Cancer Research, University of California School of Medicine, San Diego, 9500 Gilman Drive, La Jolla, CA 92093-0669, USA. [2] The FIRC Institute of Molecular Oncology Foundation, Via Adamello 16, 20139 Milan, Italy. [3] Istituto di Genetica Molecolare, Consiglio Nazionale delle Ricerche (IGM-CNR), Via Abbiategrasso 207, 27100 Pavia, Italy. [4] Departments of Medicine, University of California School of Medicine, San Diego, 9500 Gilman Drive, La Jolla, CA 92093-0669, USA. [5] Cellular and Molecular Medicine, University of California School of Medicine, San Diego, 9500 Gilman Drive, La Jolla, CA 92093-0669, USA. [6] Moores-UCSD Cancer Center, University of California School of Medicine, San Diego, 9500 Gilman Drive, La Jolla, CA 92093-0669, USA. [7] Institute of Genomic Medicine, University of California School of Medicine, San Diego, 9500 Gilman Drive, La Jolla, CA 92093-0669, USA. Correspondence and requests for materials should be addressed to R.D.K. (email: rkolodner@ucsd.edu)

Genome instability is a hallmark of cancer, and genome rearrangements are often causal mutations in human diseases[1–4]. Studies using *Saccharomyces cerevisiae* have provided insights into the genes and genetic interactions that prevent the accumulation of gross chromosomal rearrangements (GCRs), and analysis of cancer genome sequences has revealed that the corresponding pathways are affected in human cancers[5,6]. Recently, we discovered a striking set of genetic interactions between mutations in the genes encoding the Swi2/Snf2-Related chromatin remodeling complex (SWR-C) or its substrate Htz1 and mutations in the genes encoding the replication fork progression proteins Tof1 (human Timeless) and Csm3 (human Tipin) or the Mediator of the Replication Checkpoint protein Mrc1 (human Claspin)[6].

*S. cerevisiae* SWR-C (related to human SRCAP and Tip60/p400 complexes) consists of the Swr1 ATPase and 13 other subunits[7–10]. This complex mediates incorporation of the only histone variant conserved across eukaryotes, the histone H2A variant Htz1 (human H2A.Z), into chromatin by exchanging chromatin-bound H2A-H2B dimers with Htz1-H2B dimers[7,8,11]. Htz1 is enriched at gene promoters, pericentromeric chromatin, heterochromatin-euchromatin boundaries, and telomeres[12–15], and has been implicated in transcriptional regulation and preventing the spread of heterochromatin[8,16–18].

Several studies have suggested roles for SWR-C/Htz1 in maintaining genome stability. The *htz1Δ* and/or *swr1Δ* mutations cause sensitivity to DNA-damaging agents, decreased sister-chromatid cohesion, defects in recruitment of DNA double-strand breaks (DSBs) to the nuclear periphery, modest defects in resection of DSBs, a small decrease in non-homologous end joining, a small (~twofold) increase in the rate of point mutations when combined with the *pol3-L612M* mutation in DNA polymerase δ that increases base misincorporation rates, and synergistic growth interactions with mutations affecting chromosome segregation[18–26]. However, these defects are small compared to the effects caused by mutations affecting the core components of the DNA repair pathways that act in these processes.

The *htz1Δ* mutation causes a synthetic growth defect when combined with mutations in replication-related genes, including *MRC1*[11,27]. Loss of SWR-C/Htz1 also causes some phenotypes similar to those caused by loss of Mrc1, such as sensitivity to DNA-damaging agents, impaired sister chromatid cohesion, and a delay in replication progression[28–33]. During normal replication, Mrc1 couples the leading-strand DNA polymerase ε to the DNA helicase Mcm2-7[34–36] and interacts with the Tof1-Csm3 protein complex that also promotes replication fork progression; however, *mrc1* defects cause a greater reduction in the rate of replication fork progression than *csm3* or *tof1* defects[33,37–39]. Mrc1 also has other DNA replication functions: (1) maintaining stable fork pausing upon depletion of nucleotides and at natural pause sites; (2) mediating signaling of DNA replication stress by facilitating activation of the downstream Rad53 checkpoint kinase, a step that requires phosphorylation of Mrc1 by the Mec1 and Tel1 checkpoint kinases[29–31]. In contrast, Tof1-Csm3 is required for fork pausing at protein-DNA blocks and plays a less important role in checkpoint signaling[33,37,39–42]. Defects in *MRC1*, *CSM3*, and *TOF1* also cause defects in cohesion establishment, but these defects appear to define parallel pathways, one involving *MRC1* and the other involving *TOF1* and *CSM3*[43].

Here, we observe that SWR-C/Htz1 and Mrc1/Tof1/Csm3 cooperate to prevent genome instability. Increased replication stress occurs in the absence of Mrc1/Tof1/Csm3, requiring SWR-C/Htz1 to promote efficient replication progression, prevent the accumulation of aberrant replication fork structures, and prevent the accumulation of GCRs. Cells lacking Mrc1/Tof1/Csm3 and Swr1 are hypersensitive to agents that increase replication stress,

and under these conditions, HR is essential for cell survival. Together with data from structural analysis of GCRs, these results are consistent with a model wherein transiently stalled replication forks generated in the absence of Mrc1/Tof1/Csm3 are processed in the absence of SWR-C/Htz1 to unusual replication structures that lead to the accumulation of GCRs.

## Results

**Htz1 deposition by SWR-C prevents GCRs in *mrc1Δ* mutants.** In our previous study of systematically generated mutant strains, we observed that combining mutations affecting SWR-C or its substrate, Htz1, and the replication fork progression proteins Mrc1, Tof1 or Csm3 resulted in synergistic increases in dGCR rate, as assessed by semi-qualitative patch scores[6]. To understand these interactions, we performed GCR rate measurements with newly reconstructed *mrc1Δ*, *swr1Δ*, and *mrc1Δ swr1Δ* strains containing the dGCR, sGCR, or uGCR assay (Fig. 1a, b). The *mrc1Δ swr1Δ* double mutant had synergistic increases in GCR rates in all three assays, with a 5.5- to 91.8-fold increase in GCR rate relative to the higher of the two single-mutant GCR rates (Table 1; Supplementary Data 1). Moreover, deletions of genes encoding Htz1 or the non-essential subunits of SWR-C also caused synergistic increases in dGCR rates when combined with *mrc1Δ* (Fig. 1c; Table 1). The *mrc1Δ swr1Δ* and *mrc1Δ htz1Δ* double mutants and the *mrc1Δ swr1Δ htz1Δ* triple mutant had similar increases in dGCR rates (Supplementary Table 1; Supplementary Data 1), which is consistent with the hypothesis that Htz1 and SWR-C act in the same pathway, i.e., Htz1 deposition, to prevent the accumulation of GCRs in the absence of Mrc1. These experiments utilized strains constructed using gene-disruption methods; however, a subset of strains was also constructed by genetic crosses, and the resulting strains had essentially identical GCR rates (Table 1).

Defects in the genes encoding SWR-C and Htz1 alter gene expression. To evaluate whether altered gene expression possibly results in altered GCR rates, we searched the published transcriptional changes caused by deletion of *HTZ1*[16] for altered expression of the 182 genes that suppress the formation of GCRs[6], and all known DNA replication genes, as replication defects can cause increased GCR rates[44]. In the *htz1Δ* mutant, none of these genes showed increased expression, and two GCR-suppressing genes, *PHR1* and *YJL218W*, showed a minor ~ 40% reduction in expression. Defects in *PHR1* and *YJL218W* result in a minor increase in GCR patch score in the sGCR assay, which is of borderline significance, and no increase in the dGCR and uGCR assays[6]. It is formally possible that *htz1Δ* alters the expression of a GCR-suppressing gene whose expression is induced by replication stress in *mrc1Δ*, *tof1Δ,* and *csm3Δ* mutants; however, replication stress alters the expression of very few replication, repair or checkpoint genes, and none of these appears to be regulated by Htz1 under normal conditions[16,45]. Therefore it is unlikely transcriptional effects on GCR-suppressing genes can explain the increased GCR rates caused by defects in SWR-C and Htz1.

**Stable chromatin retention of Htz1 suppresses *mrc1Δ* GCRs.** Sumoylated Htz1 promotes relocalization of persistent DSBs to the nuclear periphery, and acetylated Htz1 promotes sister chromatid cohesion and the maintenance of telomere heterochromatin boundaries[21–23,46]. We tested these roles of Htz1 by eliminating the lysine residues that are the substrates for these post-translational modifications (Supplementary Table 2; Supplementary Data 1). Elimination of Htz1 sumoylation (*htz1-K(126,133)R*) did not increase the dGCR rate individually or when combined with *mrc1Δ*, *tof1Δ*, or *csm3Δ*. Elimination of Htz1

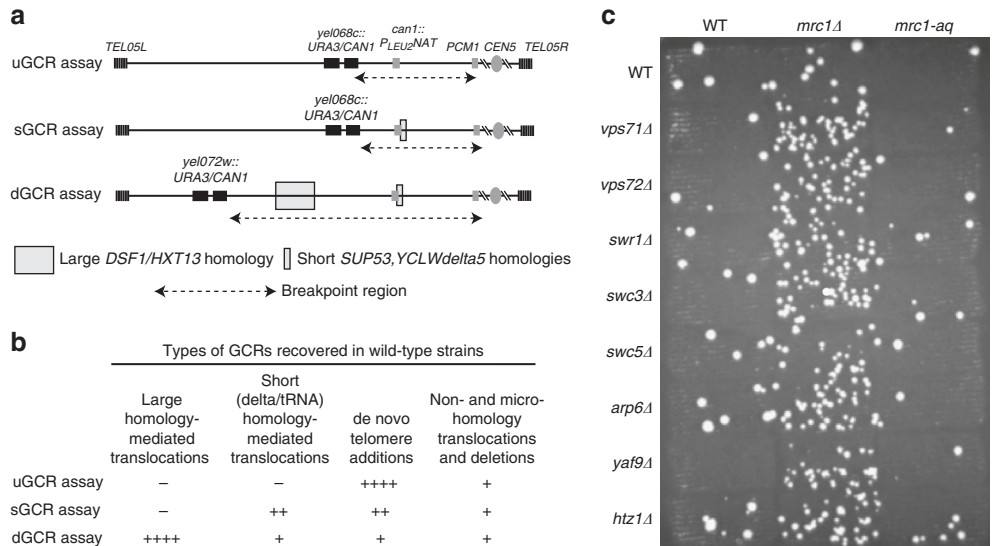

**Fig. 1** A *mrc1Δ* mutation causes increased GCRs when combined with defects in SWR-C/Htz1. **a** Genome instability was measured using three different GCR assays in which the counter-selectable genes CAN1 and URA3 were inserted as a single cassette in haploid strains at different positions in the non-essential terminal region of the left arm of chromosome V[50]. Selection against CAN1 and URA3 using the drugs canavanine (Can) and 5-fluoroorotic acid (5FOA), respectively, selects for GCRs with a breakpoint between the CAN1-URA3 cassette and the most telomeric essential gene on the left arm of chromosome V (PCM1); the DNA sequences in this breakpoint region influence the types of GCRs that are formed. In the "unique sequence" GCR (uGCR) assay, the breakpoint region contains only single-copy sequences. The "short repeated sequence" GCR (sGCR) assay contains single-copy sequences and the can1::P_LEU2-NAT locus, which introduced two short homologies that mediate GCRs by HR: SUP53, which is a 114-bp gene encoding leucine tRNA, and ~100 bp of YCLWdelta5 sequence, which has homology to the long-terminal repeats of Ty1 and Ty2 retrotransposons. The breakpoint region in the "segmental duplication" GCR (dGCR) assay contains the ~4 kb DSF1-HXT13 segmental duplication with divergent homology to regions of chromosomes IV, X, and XIV in addition to SUP53 and the YCLWdelta5 fragment. **b** The uGCR, sGCR, and dGCR assays preferentially select for different kinds of GCRs in wild-type strains[49, 50, 77, 78]. **c** Patch test for the formation of GCRs in the dGCR assay in SWR-C/Htz1 single mutants and corresponding double mutants containing the *mrc1Δ* or *mrc1-aq* mutation. Increased numbers of papillae correspond to increased GCR rates

acetylation (*htz1-K(4,9,11,15)R*) did not increase the dGCR rate individually or when combined with *mrc1Δ*; these acetylation sites were previously called K3, K8, K10, and K14[21,46]. Additionally, neither the *htz1-K(4,9,11,15,126,133)R* acetylation- and sumoylation-defective mutation nor the *htz1-K(4,9,11,15)Q* acetylation-mimic mutation caused an increased GCR rate individually or in combination with an *mrc1Δ* mutation. In contrast, C-terminal truncations of Htz1, which decrease retention of Htz1 in chromatin (*htz1—1-114* and *htz1—1-120*)[47,48], caused synergistic increases in GCR rates when combined with an *mrc1Δ* mutation. Together, these data indicate that stable chromatin retention of Htz1, but not its sumoylation or acetylation, plays a role in suppressing GCRs in the absence of Mrc1.

**SWR-C/Htz1 suppress GCRs caused by fork progression defects.** Mrc1 acts in replication fork progression and the replication checkpoint[29–32,35]. We therefore tested the separation-of-function alleles *mrc1-1-843*, an alternative version of *mrc1-c14* that confers the same replication defects (see Methods), and *mrc1-aq*, which causes a checkpoint signaling defect (Table 1; Supplementary Data 1)[30,31]. When combined with SWR-C/Htz1 defects, *mrc1-aq* did not cause an increase in the dGCR or sGCR rate and caused a small increase in the uGCR rate, which was significantly less than the effect seen with *mrc1Δ* (Table 1; Supplementary Data 1; Fig. 1c). In contrast, the *mrc1-1-843* mutation caused large synergistic increases in GCR rates in all three assays when combined with either *swr1Δ* or *htz1Δ*. Similarly, *tof1Δ* and *csm3Δ* also caused synergistic increases in GCR rates when combined with either *swr1Δ* or *htz1Δ*, but not to the extent observed with *mrc1Δ* or *mrc1-1-843* (Table 1; Supplementary Data 1). Combining *mrc1Δ* or *mrc1-1-843* with either *tof1Δ* or *csm3Δ* caused increased GCR rates, and combining *mrc1Δ* or

*mrc1-1-843* with the *tof1Δ swr1Δ* or *csm3Δ swr1Δ* double mutations caused even greater increases in GCR rates (Supplementary Table 1; Supplementary Data 1). These results suggest that mutations affecting Mrc1 (*mrc1Δ* and *mrc1-1-843*) and the Tof1-Csm3 complex (*tof1Δ* and *csm3Δ*) impair redundant replication functions, leading to increased GCR rates, which are further exacerbated in the absence of SWR-C/Htz1.

**mrc1 swr1 double mutants have an altered GCR spectrum.** To gain insights into the type of damage caused by combining defects in SWR-C/Htz1 and Mrc1, we determined the structures of the resulting GCRs. We characterized GCRs selected in the uGCR and sGCR assays, which have more informative product distributions than the homology-mediated rearrangements selected in the dGCR assay (Fig. 1b)[49]. We performed two types of analyses on these data: (1) we determined the rates of accumulation of individual classes of GCRs (observed rates) to characterize the GCR spectrum for each mutation; and, (2) for each class of GCR, we compared the observed rates to the predicted rates, which were calculated from the mutant GCR rate and the wild-type GCR spectrum (Tables 2, 3).

GCR structures were determined by analysis of paired-end whole-genome sequencing (WGS) of each parental strain and ≥ 18 or ≥ 10 independently derived GCR-containing isolates for each genotype in the uGCR (wild-type, *mrc1Δ*, *swr1Δ*, and *mrc1Δ swr1Δ*) and sGCR (wild-type, *mrc1Δ*, *mrc1-1-843*, *swr1Δ*, *mrc1Δ swr1Δ*, and *mrc1-1-843 swr1Δ*) assays, respectively. GCRs were identified using read depth, discordantly mapping read pairs, and read sequences spanning the rearrangement junctions. We identified both genomic alterations engineered into the strains, such as the *mrc1Δ* and *swr1Δ* deletions (Supplementary Figs. 1, 2), and GCR-associated genomic alterations (Supplementary

**Table 1 GCR rates of strains with defects in SWR-C/Htz1 and Mrc1/Tof1/Csm3[a]**

| Relevant genotype | dGCR assay | | | sGCR assay | | | uGCR assay | | |
|---|---|---|---|---|---|---|---|---|---|
| | RDKY | Rate[b] (x $10^{-8}$) | Fold change[c] | RDKY | Rate (x $10^{-9}$)[d] | Fold change | RDKY | Rate (x $10^{-9}$) | Fold change |
| Wild type[e] | 7635 | 8.1 [6.4-15] | 1.0 | 7964 | 6.1 [4.3-18] | 1.0 | 8625 | 1.8 [0.7-4.1] | 1.0 |
| swr1Δ[e] | 7785 | 16 [11-34] | 2.0 | 9077 | 8.4 [5.0-15] | 1.4 | 8808 | 0.8 [0.6-1.1] | 0.5 |
| htz1Δ | 8969 | 14 [10-22] | 1.7 | 9079 | 2.8 [2.2-4.4] | 0.5 | 8810 | 0.7 [0.4-1.7] | 0.4 |
| mrc1Δ[e] | 8301 | 26 [18-44] | 3.2 | 9081 | 56 [21-100] | 9.1 | 8804 | 3.0 [1.4-3.8] | 1.7 |
| mrc1Δ swr1Δ[e] | 8302 | 409 [152-964] | 50.3 | 9083 | 304 [254-402] | 49.5 | 9085 | 280 [128-514] | 155 |
| mrc1Δ htz1Δ | 8975 | 350 [269-456] | 43.1 | | n.d. | – | 9087 | 215 [102-427] | 119 |
| mrc1-aq | 8305 | 18 [12-34] | 2.2 | 9089 | 7.3 [6.4-8.7] | 1.2 | 9091 | 14 [9.1-41] | 7.6 |
| mrc1-aq swr1Δ | 8306 | 19 [12-43] | 2.3 | 9093 | 7.0 [4.4-11] | 1.1 | 9095 | 54 [36-69] | 30 |
| mrc1-aq htz1Δ | 9097 | 18 [11-55] | 2.2 | 9099 | 8.3 [5.2-10] | 1.4 | 9101 | 62 [4.3-110] | 34 |
| mrc1-1-843 | 8967 | 22 [15-30] | 2.7 | 9102 | 24 [18-47] | 3.9 | 8814 | 2.1 [1.1-4.3] | 1.2 |
| mrc1-1-843 swr1Δ | 8973 | 219 [132-341] | 27.0 | 9104 | 315 [222-444] | 51.3 | 9106 | 450 [252-623] | 250 |
| mrc1-1-843 htz1Δ | 9108 | 219 [175-316] | 26.9 | 9110 | 148 [103-342] | 24.1 | 9112 | 76 [51-156] | 42 |
| tof1Δ | 8963 | 22 [15-37] | 2.7 | 9114 | 24 [20-46] | 3.9 | 8816 | 2.4 [1.5-4.2] | 1.3 |
| tof1Δ swr1Δ[e] | 8971 | 51 [36-116] | 6.3 | 9115 | 64 [42-139] | 10 | 9117 | 12 [7.6-22] | 6.8 |
| tof1Δ htz1Δ | 9119 | 105 [48-201] | 12.9 | | n.d. | – | 9121 | 7.1 [2.3-8.5] | 3.9 |
| csm3Δ | 8965 | 34 [20-76] | 4.1 | 9128 | 25 [10-52] | 4.1 | 8806 | 5.0 [2.7-9.3] | 2.8 |
| csm3Δ swr1Δ[e] | 8972 | 72 [50-95] | 8.9 | 9130 | 54 [41-67] | 8.8 | 9132 | 7.6 [5.0-12] | 4.2 |
| csm3Δ htz1Δ | 9134 | 90 [59-202] | 11.0 | | n.d. | – | 9136 | 3.2 [1.5-5.7] | 1.8 |

[a]p values for significance calculated using the Mann-Whitney two-tailed test are presented in Supplementary Data 1.
[b]The numbers in square brackets represent the 95% confidence interval for each rate
[c]Fold change = fold change with respect to the wild-type rate for each assay
[d]n.d. not determined
[e]Rates for the corresponding dGCR strains generated by systematic crosses and haploid selection[6] are as follows (x10$^{-8}$): wild type 8.6 [4.7-13] (1), mrc1Δ 31 [14-68] (3.6), swr1Δ 13 [6.7-21] (1.5), mrc1Δ swr1Δ 396 [164-514] (46.1), tof1Δ swr1Δ 41 [28-55] (4.8), and csm3Δ swr1Δ 46 [28-194] (5.4); 95% confidence intervals and fold changes with respect to the wild type are indicated in square brackets and parentheses, respectively

**Table 2 Product distributions in the uGCR assay**

| GCR type | No. of isolates/total | Rate for GCR type[a] | Predicted rate based on wild-type product distributions[b] | Observed/predicted rate ratio |
|---|---|---|---|---|
| De novo telomere addition | | | | |
| Wild type | 15 / 20 | 1.35 [0.52-3.08] x $10^{-9}$ | 1.35 [0.52-3.08] x $10^{-9}$ | 1.00 |
| swr1Δ | 7 / 18 | 0.32 [0.23-0.43] x $10^{-9c}$ | 0.62 [0.44-0.83] x $10^{-9d}$ | 0.52 |
| mrc1Δ | 1 / 18 | 0.17 [0.08-0.21] x $10^{-9c}$ | 2.25 [1.05-2.85] x $10^{-9d}$ | 0.07 |
| mrc1Δ swr1Δ | 1 / 21 | 13.3 [6.19-24.3] x $10^{-9c}$ | 210 [97.5-383] x $10^{-9d}$ | 0.06 |
| Interstitial deletion | | | | |
| Wild type | 1 / 20 | 9.00 [3.50-20.5] x $10^{-11}$ | 9.00 [3.50-20.5] x $10^{-11}$ | 1.00 |
| swr1Δ | 2 / 18 | 9.11 [6.56-12.2] x $10^{-11}$ | 4.10 [2.95-5.50] x $10^{-11d}$ | 2.22 |
| mrc1Δ | 0 / 18 | <16.7 [7.78-21.1] x $10^{-11}$ | 15.0 [7.00-19.0] x $10^{-11}$ | 1.11 |
| mrc1Δ swr1Δ | 8 / 21 | 10,700 [4400] x $10^{-11c}$ | 1400 [650-2550] x $10^{-11d}$ | 7.62 |
| Microhomology-mediated translocation | | | | |
| Wild type | 0 / 20 | <9.00 [3.50-20.5] x $10^{-11}$ | <9.00 [3.50-20.5] x $10^{-11}$ | 1.00 |
| swr1Δ | 3 / 18 | 13.7 [9.83-18.3] x $10^{-11}$ | <4.10 [2.95-5.50] x $10^{-11d}$ | >3.33 |
| mrc1Δ | 4 / 18 | 66.7 [31.1-84.4] x $10^{-11c}$ | <15.0 [7.00-19.0] x $10^{-11d}$ | >4.44 |
| mrc1Δ swr1Δ | 9 / 21 | 12,000 [5900] x $10^{-11c}$ | <1400 [650-2500] x $10^{-11d}$ | >8.57 |
| Hairpin-mediated inverted duplication | | | | |
| Wild type | 3 / 20 | 2.70 [1.05-6.15] x $10^{-10}$ | 2.70 [1.05-6.15] x $10^{-10}$ | 1.00 |
| swr1Δ | 6 / 18 | 2.73 [1.97-3.67] x $10^{-10}$ | 1.23 [0.89-1.65] x $10^{-10d}$ | 2.22 |
| mrc1Δ | 13 / 18 | 21.7 [10.1-27.4] x $10^{-10c}$ | 4.50 [2.10-5.70] x $10^{-10d}$ | 4.81 |
| mrc1Δ swr1Δ | 3 / 21 | 400 [186-729] x $10^{-10c}$ | 420 [195-765] x $10^{-10}$ | 0.95 |

[a]Observed rate calculated by multiplying the GCR rate for each strain by the fraction of GCRs observed for a specific GCR type. 95% confidence intervals (CI) are displayed in square brackets.
[b]Predicted rate calculated by multiplying the GCR rate for each strain by the fraction of GCRs observed in the wild-type strain. 95% confidence intervals are displayed in square brackets.
[c]Rate of a specific type of GCR in the mutant strain has a 95% CI that does not overlap with the 95% CI of the wild-type rate
[d]Predicted rate based on the wild-type product distribution has a 95% CI that does not overlap with the 95% CI of the observed rate.

Figs. 3–12 and Supplementary Tables 3–5). A single GCR chromosome was identified in all GCR-containing isolates, and an otherwise normal haploid complement of chromosomes was observed in greater than 80% of the isolates analyzed (Supplementary Fig. 13).

In the uGCR assay, the GCRs in the wild-type strain were predominantly generated by de novo telomere addition (15 of 20 isolates). The observed rate of forming this class of GCRs was reduced relative to the predicted rate in the swr1Δ and mrc1Δ strains, which had uGCR rates similar to wild type, and in the

**Table 3 Product distributions in the sGCR assay**

| GCR type | No. of isolates/ total | Rate for GCR type[a] | Predicted rate based on wild-type product distributions[b] | Observed/predicted rate ratio |
|---|---|---|---|---|
| De novo telomere additions | | | | |
| Wild type | 5 / 11 | 2.77 [1.95-8.18] x $10^{-9}$ | 2.77 [1.95-8.18] x $10^{-9}$ | 1.00 |
| swr1Δ | 3 / 11 | 2.29 [1.36-4.09] x $10^{-9}$ | 3.82 [2.27-6.82] x $10^{-9}$ | 0.60 |
| mrc1Δ | 2 / 11 | 10.2 [3.82-18.2] x $10^{-9}$ | 25.5 [9.55-45.5] x $10^{-9}$ | 0.40 |
| mrc1Δ swr1Δ | 2 / 10 | 60.0 [50.0-80.0] x $10^{-9}$[c] | 136 [114-182] x $10^{-9}$[d] | 0.44 |
| mrc1-1-843 | 2 / 11 | 4.36 [3.27-8.55] x $10^{-9}$ | 10.9 [8.18-21.4] x $10^{-9}$ | 0.40 |
| mrc1-1-843 swr1Δ | 0 / 11 | <29.1 [20.0-40.0] x $10^{-9}$ | 145 [100-200] x $10^{-9}$[d] | <0.20 |
| Interstitial deletion | | | | |
| Wild type | 0 / 11 | <55.5 [39.1-164] x $10^{-11}$ | <55.5 [39.1-164] x $10^{-11}$ | 1.00 |
| swr1Δ | 0 / 11 | <70.0 [41.7-125] x $10^{-11}$ | <76.4 [45.5-136] x $10^{-11}$ | 0.92 |
| mrc1Δ | 1 / 11 | 509 [191-909] x $10^{-11}$ | <509 [191-909] x $10^{-11}$ | >1.00 |
| mrc1Δ swr1Δ | 2 / 10 | 6000 [5000-8000] x $10^{-11}$[c] | <2730 [2270-3640] x $10^{-11}$[d] | >2.20 |
| mrc1-1-843 | 0 / 11 | <200 [150-392] x $10^{-11}$ | <218 [164-427] x $10^{-11}$ | 0.92 |
| mrc1-1-843 swr1Δ | 2 / 11 | 5820 [4000-8000] x $10^{-11}$[c] | <2910 [2000-4000] x $10^{-11}$[d] | >2.00 |
| Microhomology-mediated translocation | | | | |
| Wild type | 0 / 11 | <55.5 [39.1-164] x $10^{-11}$ | <55.5 [39.1-164] x $10^{-11}$ | 1.00 |
| swr1Δ | 0 / 11 | <64.6 [38.5-115] x $10^{-11}$ | <76.4 [45.5-136] x $10^{-11}$ | 0.85 |
| mrc1Δ | 0 / 11 | <467 [175-833] x $10^{-11}$ | <509 [191-909] x $10^{-11}$ | 0.92 |
| mrc1Δ swr1Δ | 0 / 10 | <2730 [2270-3640] x $10^{-11}$ | <2730 [2270-3640] x $10^{-11}$ | 1.00 |
| mrc1-1-843 | 1 / 11 | 200 [150-392] x $10^{-11}$ | <218 [164-427] x $10^{-11}$ | >0.92 |
| mrc1-1-843 swr1Δ | 0 / 11 | <2910 [2000-4000] x $10^{-11}$ | <2910 [2000-4000] x $10^{-11}$ | 1.00 |
| Hairpin-mediated inverted duplication | | | | |
| Wild type | 0 / 11 | <5.55 [3.91-16.4] x $10^{-10}$ | <5.55 [3.91-16.4] x $10^{-10}$ | 1.00 |
| swr1Δ | 1 / 11 | 7.64 [4.55-13.6] x $10^{-10}$ | <7.64 [4.55-13.6] x $10^{-10}$ | >1.00 |
| mrc1Δ | 0 / 11 | <50.9 [19.1-90.9] x $10^{-10}$ | <50.9 [19.1-90.9] x $10^{-10}$ | 1.00 |
| mrc1Δ swr1Δ | 1 / 10 | 300 [250-400] x $10^{-10}$[c] | <273 [227-364] x $10^{-10}$ | >1.10 |
| mrc1-1-843 | 0 / 11 | <21.8 [16.4-42.7] x $10^{-10}$ | <21.8 [16.4-42.7] x $10^{-10}$ | 1.00 |
| mrc1-1-843 swr1Δ | 6 / 11 | 1,750 [1400] x $10^{-10}$[c] | <291 [200-400] x $10^{-10}$[d] | >6.00 |
| Homology-mediated inverted duplication | | | | |
| Wild type | 0 / 11 | <5.55 [3.91-16.4] x $10^{-10}$ | <5.55 [3.91-16.4] x $10^{-10}$ | 1.00 |
| swr1Δ | 0 / 11 | <7.64 [4.55-13.6] x $10^{-10}$ | <7.64 [4.55-13.6] x $10^{-10}$ | 1.00 |
| mrc1Δ | 0 / 11 | <50.9 [19.1-90.9] x $10^{-10}$ | <50.9 [19.1-90.9] x $10^{-10}$ | 1.00 |
| mrc1Δ swr1Δ | 0 / 10 | <300 [250-400] x $10^{-10}$ | <300 [250-400] x $10^{-10}$ | 1.00 |
| mrc1-1-843 | 1 / 11 | 21.8 [16.4-42.7] x $10^{-10}$* | <21.8 [16.4-42.7] x $10^{-10}$ | >1.00 |
| mrc1-1-843 swr1Δ | 0 / 11 | <291 [200-400] x $10^{-10}$ | <291 [200-400] x $10^{-10}$ | 1.00 |
| Homology-mediated translocation | | | | |
| Wild type | 6 / 11 | 3.33 [2.35-9.82] x $10^{-9}$ | 3.33 [2.35-9.82] x $10^{-9}$ | 1.00 |
| swr1Δ | 3 / 11 | 2.10 [1.25-3.75] x $10^{-9}$ | 4.58 [2.73-8.18] x $10^{-9}$ | 0.46 |
| mrc1Δ | 8 / 11 | 40.7 [15.3-72.7] x $10^{-9}$[c] | 30.5 [11.5-54.5] x $10^{-9}$ | 1.33 |
| mrc1Δ swr1Δ | 5 / 10 | 150 [125-200] x $10^{-9}$[c] | 164 [136-218] x $10^{-9}$ | 0.92 |
| mrc1-1-843 | 8 / 11 | 16.0 [12.0-31.3] x $10^{-9}$[c] | 13.1 [9.82-25.6] x $10^{-9}$ | 1.22 |
| mrc1-1-843 swr1Δ | 3 / 11 | 87.3 [60.0-120] x $10^{-9}$[c] | 175 [120-240] x $10^{-9}$[d] | 0.50 |

[a]Observed rate calculated by multiplying the GCR rates by the fraction of GCRs observed of a specific GCR type. 95% confidence intervals (CI) are displayed in square brackets.
[b]Predicted rate calculated by multiplying the GCR rate for each strain by the fraction of GCRs observed in the wild-type strain. 95% confidence intervals are displayed in square brackets.
[c]Rate of a specific type of GCR in the mutant strain has a 95% CI that does not overlap with the 95% CI of the wild-type rate
[d]Predicted rate based on the wild-type product distribution has a 95% CI that does not overlap with the 95% CI of the observed rate.

mrc1Δ swr1Δ strain, which had a 156-fold increase in GCR rate (Table 2). In contrast, the rates of formation of interstitial deletions, microhomology-mediated translocations, and hairpin-mediated inverted duplications were generally increased in the mutant strains, except for interstitial deletions in the mrc1Δ strain and hairpin-mediated inverted duplications in the mrc1Δ swr1Δ strain (Table 2).

In the sGCR assay, the GCRs included those observed in the uGCR assay and rearrangements mediated by the short tRNA gene and delta fragment homologies introduced at the can1:: $P_{LEU2}$-NAT locus[50]. Like the uGCR assay, the rate of accumulation of de novo telomere addition GCRs was reduced relative to the rate predicted from the wild-type spectrum (Table 3)[6,50];

however, unlike the uGCR assay, the predicted and observed rates had partially overlapping 95% confidence intervals except for the rates of the mrc1Δ swr1Δ and mrc1-1-843 swr1Δ double mutants. Additionally, the rate of accumulation of interstitial deletions was increased in the mrc1Δ swr1Δ and mrc1-1-843 swr1Δ double mutant strains, the rate of hairpin-mediated inverted duplications was increased in the mrc1-1-843 swr1Δ double mutant strain, and the rate of homology-mediated translocations was reduced in the mrc1-1-843 swr1Δ double mutant strain relative to the predicted rates (Table 3).

The changes in the distributions of GCRs in mutant strains are best understood in terms of a modest bias against the accumulation of de novo telomere addition GCRs, which then

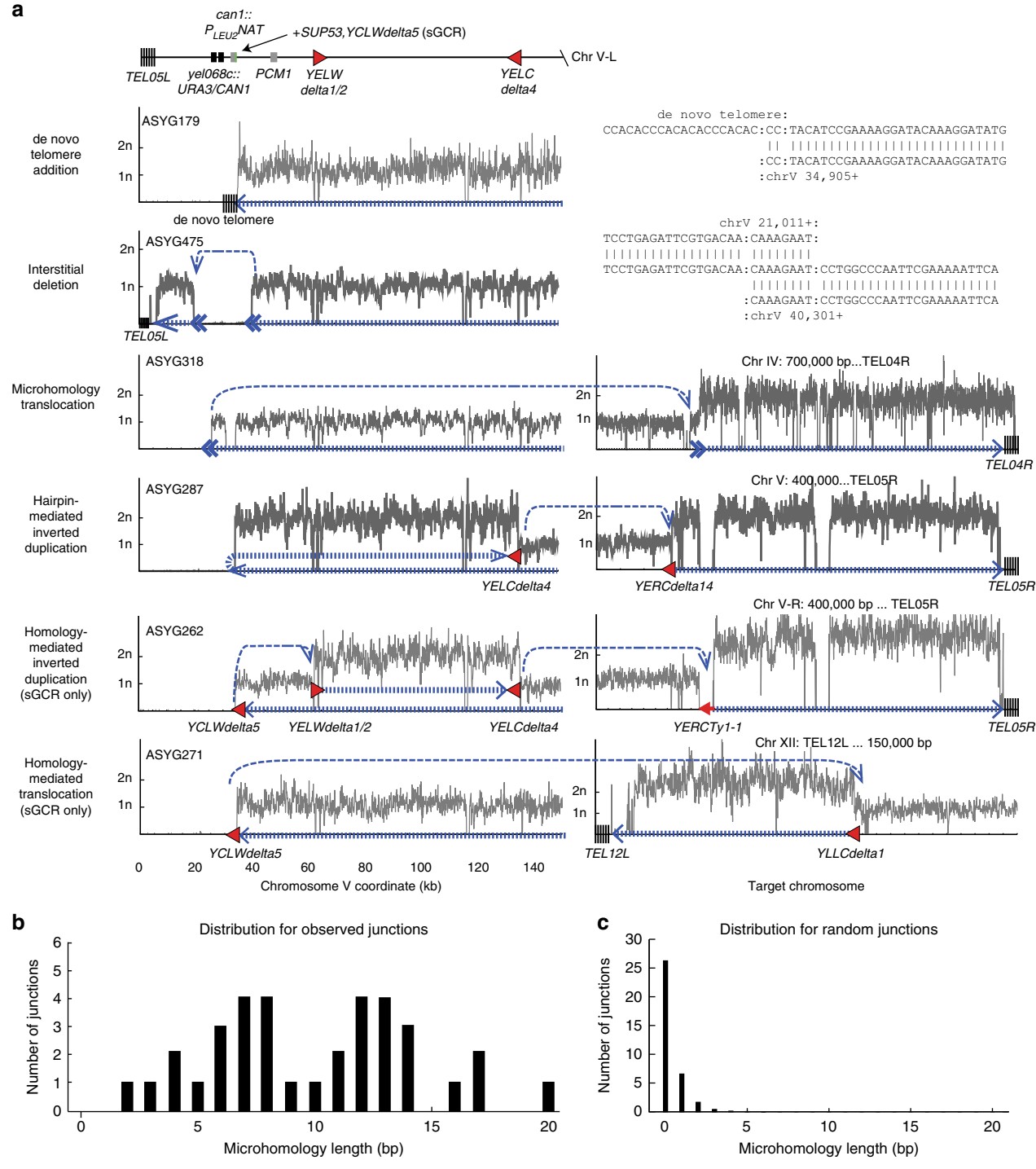

**b** Distribution for observed junctions

**c** Distribution for random junctions

leads to the formation of other types of GCRs. The changes in the sGCR assay were subtler than in the uGCR assay due to the ability of the sGCR assay to select for short homology-mediated translocations in most strains. Remarkably, the length of homology at the microhomology-mediated translocation and interstitial deletion breakpoints in these strains were longer (range 2–20 bp, median 10.0 bp; Fig. 2b) than would be expected for random junctions (Fig. 2c[51]), suggesting that these GCRs are generated by microhomology-mediated end joining or a form of HR that acts on short sequences[51]. These results suggest that SWR-C/Htz1 plays an important role in preventing the formation of DNA damage and/or preventing DNA damage from being channeled into GCRs, but only plays a minor role in influencing

the selection of DNA repair pathways that act to generate GCRs[5,44].

**SWR-C/Htz1 defects do not cause checkpoint activation**. The synergistic increases in GCR rates in strains with combined defects in Mrc1/Tof1/Csm3 and SWR-C/Htz1 could be due to increased levels of DNA damage or altered processing of damage caused by one of the mutations. We therefore used two assays to evaluate checkpoint activation and DNA damage formation in vivo (Fig. 3 and Supplementary Fig. 14): (1) cytological analysis of Ddc2-GFP foci (Fig. 3a, b), which reflects activation of the Mec1 checkpoint kinase[52]; and (2) FACS analysis of expression of

**Fig. 2** Structures of GCRs isolated in strains with defects in *MRC1* and *SWR1*. **a** Copy number analysis of chrV L (left) and the target chromosomes (right) for representative GCRs based on whole-genome sequencing. The thick hashed blue arrow indicates sequences within the GCR; the thin dashed blue arrow indicates connectivity between portions of the GCR that map to different regions of the reference chromosome(s). Filled triangles are Ty-related (red) multi-copy sequences involved in GCR-related HR events. Junction sequences are displayed for rearrangements not associated with copy number increases. Telomere addition GCRs had deletion of the terminal region of chromosome V, including the *CAN1-URA3* cassette, and addition of a de novo telomere to the broken end; the junctions involved short telomere-like sequences on chromosome V. Interstitial deletions spanned the *CAN1-URA3* cassette and contained microhomology at the deletion junctions. Hairpin-mediated GCRs had deletion of the terminal region of chromosome V and hairpin formation, followed by inverted duplication of a region of chromosome V ending in a region of homology, either a *Ty delta* sequence or *PAU* gene, and a subsequent secondary translocation with a homologous target elsewhere in the genome. In all hairpin-mediated inverted duplication GCRs, subsequent rearrangements involved a non-reciprocal translocation with a target chromosome, involving duplication of the target chromosome from the targeted homology to the telomere. Homology-mediated inverted duplications are similar to hairpin-mediated inverted duplications except that the fold-back loop is formed by HR between the *YCLWdelta5* fragment in *can1::P_LEU2-NAT* and other *Ty delta* sequences on the left arm of chromosome V, leading to inverted duplication of the flanking sequence. **b** Distribution of the microhomology lengths at the junctions in the 35 microhomology-mediated translocations and interstitial deletions observed in the uGCR and sGCR products from the wild-type strain, the *mrc1Δ, swr1Δ*, and *mrc1-1-843* single mutants, and the *mrc1Δ swr1Δ* and *mrc1-1-843 swr1Δ* double mutants. **c** Distribution of the breakpoint junction lengths for randomly generated translocations scaled so that the total number of events was 35

Hug1-GFP (Fig. 3c, d), which is a downstream marker for checkpoint activation[53]. The *mrc1Δ* and *mrc1-1-843* mutations, but not *mrc1-aq*, caused increased levels of Ddc2 foci and Hug1-GFP expression. In contrast, *swr1Δ* and *htz1Δ* did not increase the levels of Ddc2 foci and Hug1-GFP expression individually or when combined with *mrc1* mutations, although the frequency of Ddc2 foci was slightly decreased in the *mrc1Δ htz1Δ* double mutant. The *tof1Δ* and *csm3Δ* mutations behaved similarly to *mrc1Δ* and *mrc1-1-843*, although they caused lower levels of checkpoint activation (Supplementary Fig. 14).

To determine if the increased levels of Ddc2 foci observed in the *mrc1Δ, tof1Δ*, and *csm3Δ* single mutants and the *mrc1Δ swr1Δ* and *mrc1Δ htz1Δ* double mutants were the result of increased checkpoint activation rather than an altered cell cycle distribution, we analyzed the distribution of non-budded and small-medium budded cells in cultures of these mutants and determined the fraction of the cell types that contained Ddc2-GFP foci, which were only found in small-medium budded cells. The *mrc1Δ, tof1Δ*, and *csm3Δ* mutants had a small increase in the frequency of small-medium budded cells and an increased frequency of Ddc2-GFP foci among the small-medium budded cells, whereas the *swr1Δ* and *htz1Δ* single mutations caused little or no effect (% small-medium budded cells/% small-medium budded cells containing Ddc2-GFP foci: wild type 33.6/9.2%; *swr1Δ* 28.8/8.6%; *htz1Δ* 31.1/12.1%; *mrc1Δ* 47.7/32.7%; *tof1Δ* 36.9/34.1%; and *csm3Δ* 34.8/40.3%). Similar analysis of the *mrc1Δ swr1Δ* and *mrc1Δ htz1Δ* double mutants showed that they had a small increase in the frequency of Ddc2-GFP foci among the small-medium budded cells compared to the *mrc1Δ* single mutant (*mrc1Δ swr1Δ* 42.5/42.2%; *mrc1Δ htz1Δ* 27.4/35.8%). These results were consistent with the results from total cell counts (Fig. 3a, b). Overall, these results indicate that *mrc1Δ, tof1Δ*, and *csm3Δ* mutations, but not *swr1Δ* and *htz1Δ*, cause increased levels of DNA damage, resulting in constitutive checkpoint activation. Therefore, loss of SWR-C/Htz1 does not increase the steady state level of DNA damage in cells lacking Mrc1/Tof1/Csm3, but the existing damage is mis-repaired at higher rates in the absence of SWR-C/Htz1, leading to the formation of GCRs, suggesting that SWR-C/Htz1 is required for accurate processing of the damage.

**HR repairs DNA damage in strains without Mrc1 and SWR-C/Htz1**. Although the *mrc1Δ htz1Δ* and *mrc1Δ swr1Δ* double mutants had modest or no increases in checkpoint activation compared to the *mrc1Δ* single mutant, the *mrc1Δ htz1Δ* and *mrc1Δ swr1Δ* double mutants had slower doubling times than any of the respective single mutants (Fig. 3e), which is consistent with defects in resolving the DNA damage caused by *mrc1Δ, tof1Δ*, or

*csm3Δ* mutations. Sister chromatid HR is a major mechanism for repairing DNA damage in *S. cerevisiae* without generating GCRs[5]. To test if HR acts to repair the damage caused by *mrc1Δ* in strains without SWR-C/Htz1, we measured the doubling times of wild-type, *mrc1Δ, swr1Δ*, and *mrc1Δ swr1Δ* strains with and without a *rad52Δ* mutation, which causes a substantial HR defect[54]. The *rad52Δ* mutation caused a modest increase in doubling time in the wild-type, *mrc1Δ*, and *swr1Δ* strains and a severely prolonged doubling time in the *mrc1Δ swr1Δ* strain (Fig. 3f). Therefore in the absence of SWR-C/Htz1, HR plays a crucial role in repairing the DNA damage that occurs in *mrc1Δ* strains.

**Replication stress sensitivity without Mrc1 and SWR-C/Htz1**. To gain insight into the role of the Mrc1/Tof1/Csm3-SWR-C/Htz1 interaction during replication stress as opposed to unperturbed cell growth, we measured the sensitivity of various single and double mutants to methyl methane sulfonate (MMS) and hydroxyurea (HU), which block DNA replication by causing DNA alkylation damage and depleting dNTP pools, respectively (Fig. 4a, b, and Supplementary Fig. 15)[55,56]. The *swr1Δ* and *htz1Δ* single mutants were weakly sensitive to MMS (*htz1Δ* > *swr1Δ*) and HU (*htz1Δ* >> *swr1Δ*), consistent with previous studies[18]. The *mrc1Δ* and *mrc1-1-843* mutations individually caused weak or no sensitivity, respectively, to MMS and HU, but caused strikingly increased sensitivity to both drugs when combined with either *swr1Δ* or *htz1Δ*. In contrast, *mrc1-aq* did not cause increased HU or MMS sensitivity either as a single mutation or in combination with either *swr1Δ* or *htz1Δ*. The *tof1Δ* and *csm3Δ* mutations caused similar but weaker patterns of sensitivity to MMS and HU compared to *mrc1Δ* (Supplementary Fig. 15). In addition, the *rad52Δ mrc1Δ* and *rad52Δ swr1Δ* double mutants had synergistically increased sensitivity to HU compared to the *rad52Δ* single mutant (*rad52Δ mrc1Δ* > *rad52Δ swr1Δ*) (Fig. 4c). Consistent with the effect of *rad52Δ* on the doubling time of the *mrc1Δ swr1Δ* strain (Fig. 3f), the *rad52Δ mrc1Δ swr1Δ* triple mutant was extremely sensitive to HU and unable to grow even at the lowest HU concentration tested (10 mM) (Fig. 4c). Thus, the combination of defects in Mrc1/Tof1/Csm3 and SWR-C/Htz1 causes hypersensitivity to replication stress induced by exogenous agents and a requirement for HR for survival under these conditions.

**Cell cycle defects without Mrc1 and SWR-C upon HU treatment**. Plate-based HU and MMS hypersensitivity reflects the effect of replication stress over the course of 25–40 rounds of cell division. We next investigated if this sensitivity causes defects

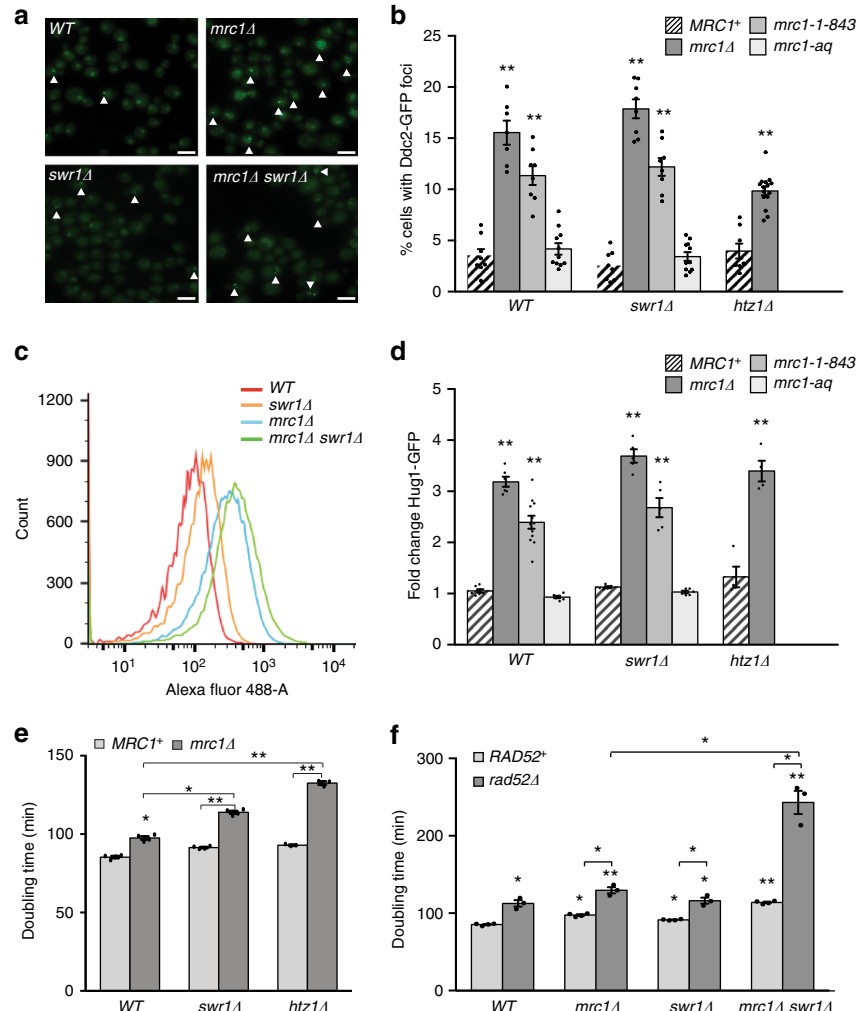

**Fig 3** s*wr1Δ/htz1Δ* mutations synergistically increase growth defects but not checkpoint activation in combination with a *mrc1Δ* mutation. **a** Ddc2-GFP foci in the indicated strains. White triangles indicate Ddc2-GFP foci. Scale bar, 10 μM. **b**. Distribution of Ddc2-GFP foci in strains with various *mrc1* mutations with and without an additional *swr1Δ* or *htz1Δ* mutation. The average percentage of cells with Ddc2-GFP foci was calculated from at least eight images with ~200 cells each, from at least two independent experiments. The error bars represent the standard error of the mean. **c** FACS-based measurement of Hug1-GFP induction in individual cells can be used to visualize the relative Hug1-GFP induction for cell populations. **d** Hug1-GFP induction in strains with various *mrc1* mutations with or without the *swr1Δ* or *htz1Δ* mutation. The fold changes were calculated by dividing the average Hug1-GFP level for each strain by the average Hug1-GFP level of the wild-type strain measured by FACS in the same experiment. A minimum of four and a maximum of 12 independent cultures derived from a minimum of two independent strain isolates for each genotype were analyzed. The mean fold changes were calculated, and the error bars represent the standard error of the mean. **e**, **f** Doubling times of the indicated strains. The mean doubling time and standard error were calculated from the doubling times of a minimum of three independent cultures derived from a minimum of two independent isolates of each genotype. In (**b**, **d**, **e**, **f**), individual observations are shown as dots overlaid on the bar graphs, and asterisks represent significant differences with respect to the wild-type strain (unless otherwise indicated) as follows: * $p < 0.005$, ** $p < 0.0005$ (two-tailed *t*-test)

during the progression of a single cell cycle. Cells were first arrested in G1 phase, then allowed to progress to mid-S phase in medium containing 200 mM HU, and then released into medium lacking HU. We assessed cell cycle recovery by monitoring DNA content by FACS analysis at different time points (Supplementary Fig. 16). The *swr1Δ* and *htz1Δ* single mutants displayed slowed S phase progression in the presence of HU but recovered upon HU release (Supplementary Fig. 16). The *tof1Δ* single mutant had a similar profile to wild-type cells; however, the *tof1Δ swr1Δ* and *tof1Δ htz1Δ* double mutants showed little progression into S-phase in the presence of HU and were highly defective in recovering from HU treatment. Compared to the *tof1Δ* strain, the *mrc1Δ* strain had a greater defect in S phase progression in the presence of HU, and because of the severity of this defect, the differences between the *mrc1Δ* single mutant and the *mrc1Δ*

*swr1Δ* and *mrc1Δ htz1Δ* double mutants were less striking than between the corresponding *tof1Δ* single and double mutants (red curves in Supplementary Fig. 16). Upon release from HU, the *mrc1Δ swr1Δ* and *mrc1Δ htz1Δ* double mutants showed more severe replication defects compared to the *mrc1Δ* single mutant. Overall, the effect of *swr1Δ* was similar to but slightly more severe than that of *htz1Δ*, both as single mutations and when combined with *tof1Δ* or *mrc1Δ*. Together these data suggest that loss of Mrc1/Tof1/Csm3 in combination with loss of SWR-C/Htz1 leads to defects in replication during recovery from replication stress.

**Fork defects without Mrc1/Tof1 and SWR-C upon HU treatment.** To better characterize the DNA replication defects caused by absence of Mrc1/Tof1/Csm3 and SWR-C/Htz1, we used two-

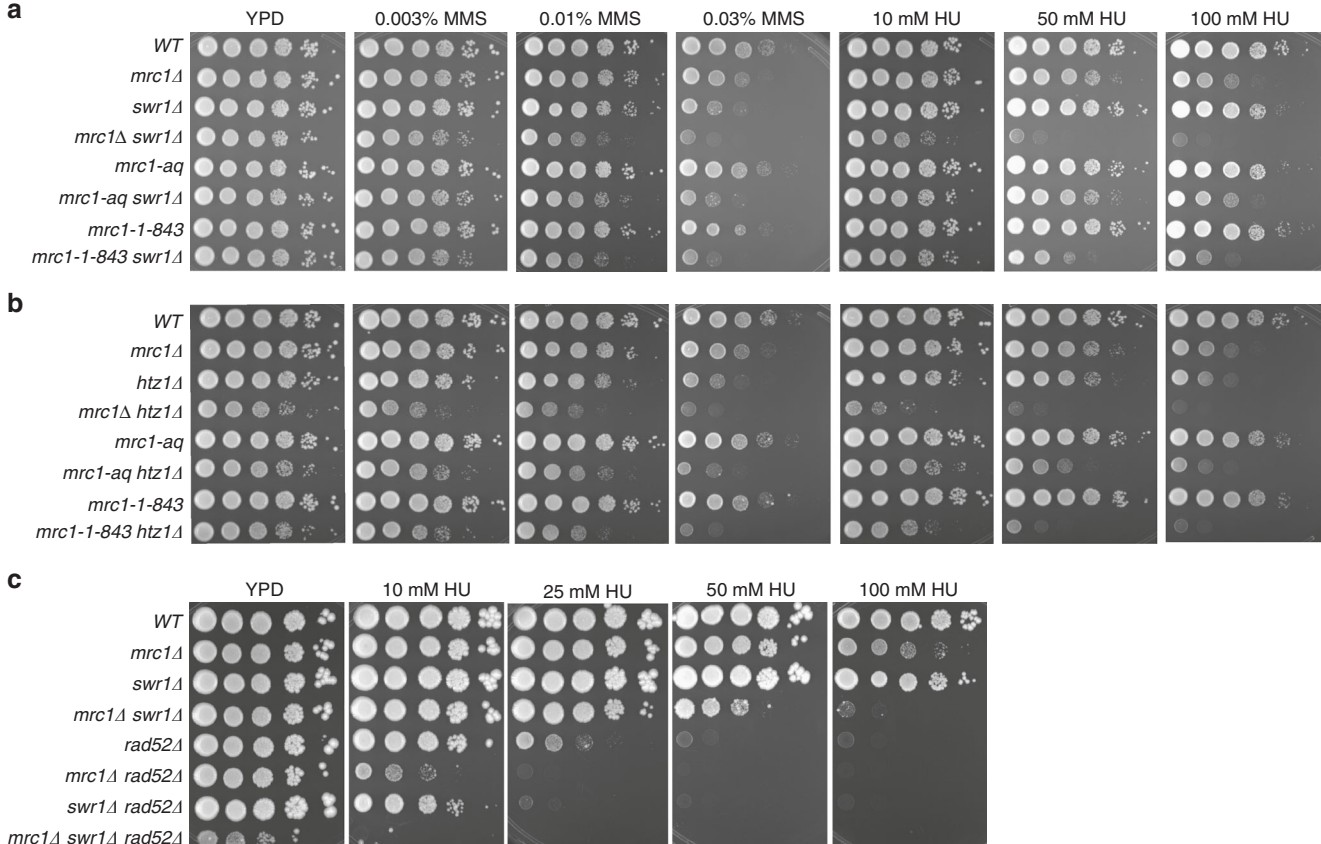

**Fig. 4** Defects in Mrc1, SWR-C/Htz1, and Rad52 cause synergistic sensitivity to DNA replication stress. **a**, **b** Tenfold serial dilutions of wild-type (WT) and mutant strains containing the *mrc1Δ*, *mrc1-aq*, or *mrc1-1-843* mutation with or without the *swr1Δ* mutation (**a**) or *htz1Δ* mutation (**b**) were plated on media containing 0–0.03% MMS or 0–100 mM HU to test their sensitivity to replication stress. Plates were imaged after 2 days of growth at 30 °C. **c** Tenfold serial dilutions of wild-type and mutant strains containing the *mrc1Δ*, *swr1Δ*, or *mrc1Δ swr1Δ* mutations with or without the *rad52Δ* mutation were plated on media containing 0–100 mM HU to test their sensitivity to replication stress. Plates were imaged after 6 days of growth at 30 °C because of the severe growth defect of *rad52Δ*-containing strains

dimensional (2D) gel electrophoresis to analyze the profile of replication intermediates adjacent to the early origin of replication *ARS305* in cells replicating in the presence of 200 mM HU (Fig. 5a). After 1 h of replication in HU, the predominant replication intermediates in wild-type cells were replication bubbles formed during origin firing and large Y-arcs representing replication forks in the fragment analyzed; these intermediates showed decreased intensities at later time points, consistent with progression of the replication forks beyond the fragment analyzed (Fig. 5b). A population of X-shaped molecules was also visible in wild-type cells 2 h after release into HU; these likely represent hemicatenanes and/or transient recombination structures at stalled replication forks (Fig. 5b)[57–59].

The *swr1Δ* and *htz1Δ* single mutants had persistent bubbles and Y-arc molecules (Fig. 5b; for example, see the 2-h time point), indicating slowed replication. This effect was accompanied by and potentially caused by reduced origin firing efficiency, as indicated by the reduced intensity of the bubble arc. These results are consistent with the observed cell cycle profiles (Supplementary Fig. 16A) and explain the slight delay in S phase progression (Fig. 5b, compare the 4 h FACS time points).

The *mrc1Δ* and *tof1Δ* single mutants had more pronounced replication defects than the *swr1Δ* and *htz1Δ* single mutants: the intensity of the bubble arc was reduced (Fig. 5b and Supplementary Fig. 17), and Y-arc and X-molecules accumulated at later time points (Supplementary Figs. 17, 18), suggesting differential processing of the stalled replication fork intermediates to

recombination-like structures. Notably, small Y-molecules and cone structures were observed in *mrc1Δ* cells, resembling those reported for *rad53* checkpoint mutants replicating in the presence of HU[60,61], where the structures were attributed to resection of stalled forks and formation of reversed forks[61–64] (Supplementary Fig. 17).

The *tof1Δ swr1Δ* and *tof1Δ htz1Δ* mutants had greatly exacerbated replication defects with substantially reduced origin firing, persistence of replication intermediates, and accumulation of cone structures (Fig. 5b and Supplementary Fig. 18). In contrast, unusual replication intermediates did not accumulate at detectable levels in the *mrc1Δ swr1Δ* and *mrc1Δ htz1Δ* double mutants; however, this likely reflects the extreme replication defects of these double mutants, as evidenced by both the low intensity of the bubble arc and limited progression into S-phase as measured by FACS (Supplementary Fig. 17- compare to Fig. 5b; Supplementary Fig. 16).

Because the aberrant DNA replication intermediates were more easily observable in the *tof1Δ swr1Δ* and *tof1Δ htz1Δ* double mutants, we tested whether these would be resolved if the strains were allowed to recover from a transient HU block. To test this, cells were first synchronized in G1 phase, then released into 200 mM HU for 4 h, and finally released into medium without HU for 2 h. The wild-type, *swr1Δ, htz1Δ*, and *tof1Δ* strains were able to recover from HU treatment over the course of 2 h, although some persistent replication intermediates were still observed in *swr1Δ*, *htz1Δ*, and *tof1Δ* cells during the first hour of recovery

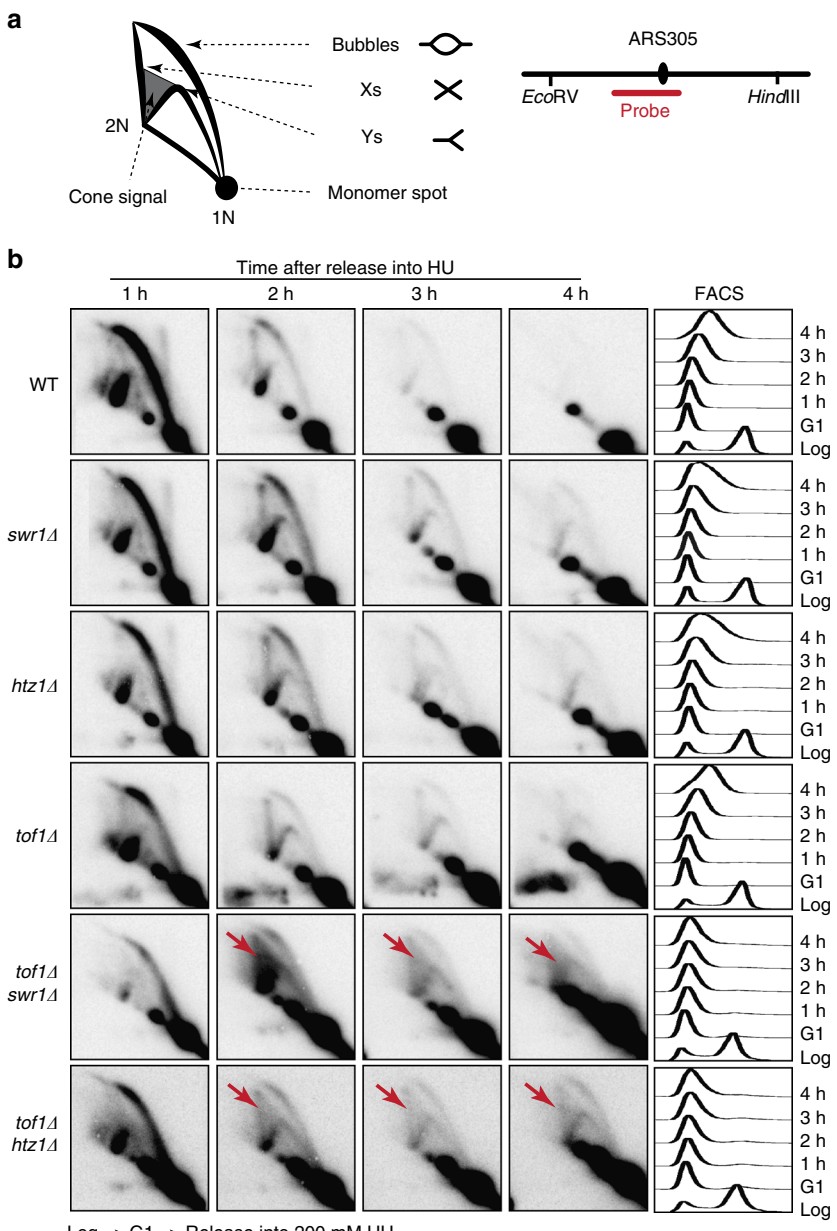

**Fig. 5** Replication defects in *tof1Δ swr1Δ* and *tof1Δ htz1Δ* strains in the presence of HU. **a** Left: migration patterns of different types of replication structures observed by 2-D gel electrophoresis. Right: diagram of the hybridization probe used to monitor replication structures near the early origin *ARS305*. **b** 2D gels and FACS profiles of the indicated strains released from G1 arrest and replicating in the presence of 200 mM HU. Red arrows indicate the accumulation of cone structures in the double mutants

(Supplementary Fig. 18). In contrast, the *tof1Δ swr1Δ* and *tof1Δ htz1Δ* double mutants were more defective in cell cycle progression than the single mutants, and the levels of cone structures forming in the *ARS305* region decreased over the 2-h time course as they were converted to other structures or DSBs (Supplementary Fig. 18). Together, these results indicate that SWR-C/Htz1 helps promote the processing of stalled DNA replication fork intermediates that are induced by HU and defects in Mrc1 and Tof1, facilitating replication progression.

## Discussion
Previous studies have suggested a role for SWR-C/Htz1 in genome maintenance; however, the magnitude of the defects in genome maintenance-related functions caused by mutations

affecting SWR-C and Htz1 are small[18,20–26,65]. Here, we found that loss of SWR-C/Htz1 plays a profound role in genome maintenance in the presence of replication defects caused by *mrc1Δ*, *mrc1-1-843*, *tof1Δ*, or *csm3Δ*. These combined defects caused: (1) significant increases in GCR rates and modest changes in the spectrum of GCRs formed, (2) hypersensitivity to chemical agents that induce replication stress, (3) a requirement for Rad52 for promoting cell division and survival under conditions of replicative stress, and (4) severe HU-induced replication defects leading to increased levels of aberrant replication fork structures. The *mrc1Δ*, *tof1Δ*, and *csm3Δ* mutations caused constitutive activation of the DNA damage checkpoint response, but this was not exacerbated by additional mutations in *SWR1* or *HTZ1*. Together, these data support a model in which the absence of functional Mrc1 or Tof1-Csm3 causes replication defects

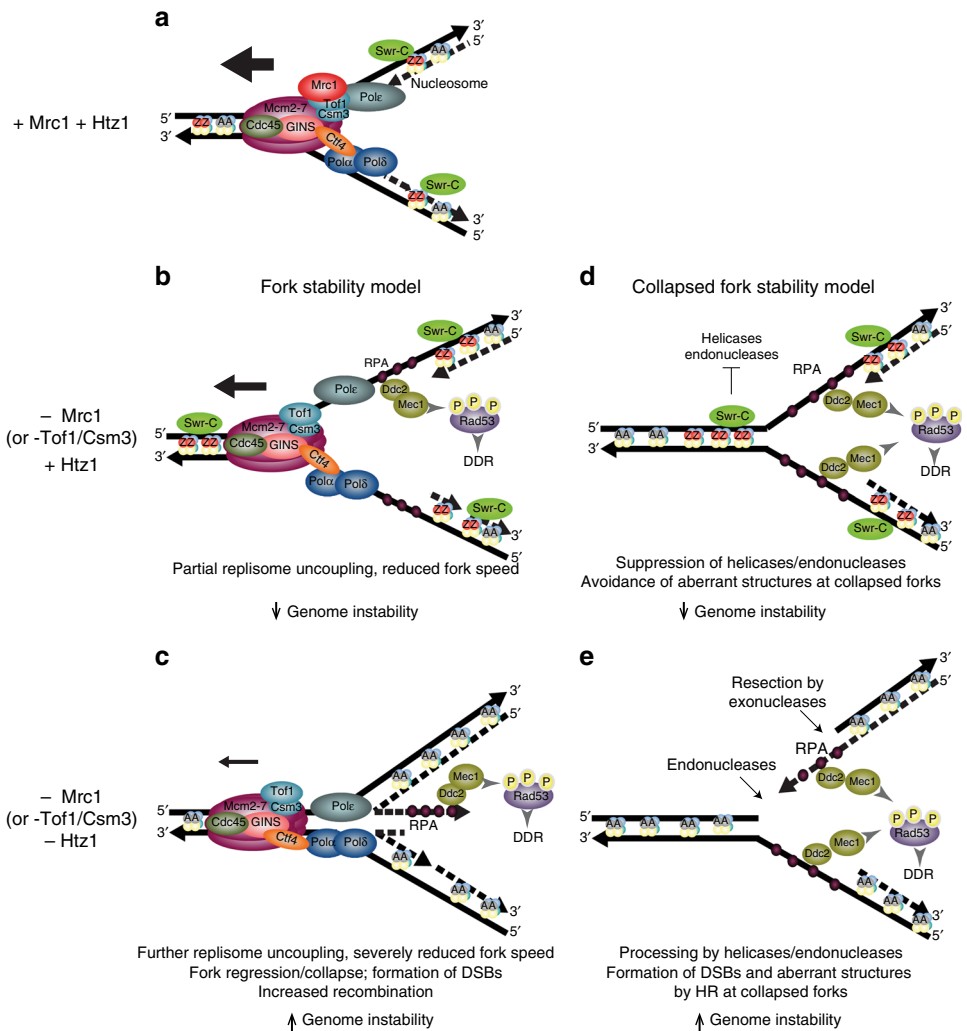

**Fig. 6** Hypotheses for the role of Htz1 in preventing genome instability. **a** Schematic of a normal replication fork showing key components including the Cdc45-Mcm2–7-GINS helicase complex, the leading (Polε) and lagging strand (Polα and Polδ) DNA polymerases, Ctf4, which couples the helicase and Polα, and Mrc1, which couples the helicase and Polε and interacts with Tof1-Csm3. Other replisome components are omitted for clarity. Also shown are nucleosomes containing the canonical histone H2A (grey circles) or the histone variant H2A.Z/Htz1 (red circles), which is incorporated by the SWR-C. **b** (Fork Stability Model) In the absence of Mrc1, replication fork progression is slowed and replisome uncoupling can occur during replication stress. The DNA damage response (DDR) is constitutively induced. The presence of Htz1 in normal chromatin prevents further replisome uncoupling or fork regression and collapse, preventing genome instability. The absence of Tof1 or Csm3 leads to similar albeit weaker defects. **c** In the absence of Mrc1 and Htz1, replication fork progression is severely impeded. Although DDR signaling is not elevated relative to the loss of Mrc1 alone, abnormal replication intermediates are formed, which are processed into DSBs. HR repairs the DNA damage and is required for survival during replication stress. Repair of the DNA damage using non-allelic targets leads to elevated genome instability. **d** (Collapsed Fork Stability Model) In the absence of Mrc1, increased levels of collapsed forks are formed. Remaining single-stranded DNA regions induce checkpoint activation, but incorporation of Htz1 near the sites of damage suppresses Rad51 filament formation and other HR-mediated repair processes, allowing the collapsed fork to be repaired by approach of an oppositely oriented fork. **e** In the absence of Mrc1 and Htz1, the collapsed fork is subjected to processing by structure-specific helicases and endonucleases, resulting in DSBs and other substrates for intermolecular HR, which generate aberrant replication fork structures and GCRs

resulting in DNA damage, necessitating chromatin-bound Htz1 for replication fork dynamics to prevent genome instability (Fig. 6).

Mrc1/Tof1/Csm3 are involved in normal replication fork progression and replication stress signaling. The *mrc1Δ* and *tof1Δ* mutants have increased levels of aberrant replication fork structures, and *mrc1Δ*, *tof1Δ*, and *csm3Δ* mutants also have increased DNA damage checkpoint activation. Although Mrc1, Tof1, and Csm3 have similar roles at the replication fork, mutations affecting *MRC1*, *TOF1*, and *CSM3* cause distinct phenotypes: (1) Tof1-Csm3 promotes the Mrc1-replisome association, but the converse is not true[38]; (2) loss of Mrc1 causes a more severe reduction in replication fork speed than the absence of Tof1-

Csm3[33,37–39]; (3) Tof1-Csm3 is required for fork pausing at protein-DNA blocks, whereas Mrc1 is not[37,39,40]; and (4) Mrc1 and Tof1-Csm3 function in distinct pathways that promote the establishment of sister chromatid cohesion, albeit not to the same extent as cohesin itself[43]. Consistent with this, we found that: (1) *mrc1Δ* and the replication-defective *mrc1-1-843* mutation cause large synergistic increases in GCR rates in combination with *tof1Δ*, *csm3Δ*, *swr1Δ*, or *htz1Δ* mutations, and in triple mutants containing *tof1Δ* or *csm3Δ* and *swr1Δ* or *htz1Δ*; and (2) *mrc1Δ* causes higher GCR rates than *tof1Δ* or *csm3Δ* when combined with *swr1Δ* or *htz1Δ*. Together with the observation that the checkpoint-defective *mrc1-aq* mutation caused much lower or none of the defects described above, depending on the assay, these

results argue for an important role of SWR-C/Htz1 in suppressing GCRs caused by defective DNA replication. The effects observed here appear to be distinct from the ~twofold increase in the rate of point mutations when defects in SWR-C/Htz1 are combined with the *pol3-L612M* mutation in DNA polymerase δ, which increases base misincorporation rates[24].

How does SWR-C/Htz1 promote the repair of replication damage? The similar dGCR rates of the *mrc1Δ swr1Δ htz1Δ* triple mutant and the *mrc1Δ swr1Δ* and *mrc1Δ htz1Δ* double mutants indicate that SWR-C suppresses GCRs by incorporating Htz1 into chromatin. Analysis of separation-of-function alleles of *HTZ1* indicates that the known roles of Htz1 in promoting the relocalization of persistent DSBs to the nuclear periphery, promoting sister chromatid cohesion, and maintaining telomeric heterochromatin boundaries do not suppress the formation of GCRs in the absence of Mrc1. Moreover, the roles of SWR-C/Htz1 in preventing the spread of heterochromatin at silenced regions[7,16] are unlikely to explain the effects observed here: (1) only 2 of ~ 240 genes involved in DNA replication or genome stability show modestly (~40%) reduced expression in the *htz1Δ* mutant, and these two genes only play a minor, if any, role in suppressing GCRs; (2) replication stress alters the expression of very few replication, repair or checkpoint genes[45], and none of these appears to be regulated by Htz1 under normal conditions; and (3) the aberrant replication structures in strains with mutations in *MRC1/TOF1* and *SWR1/HTZ1* were observed at *ARS305*, which does not correspond to an Htz1-activated domain[16]. Finally, C-terminal truncation mutations of *HTZ1* encoding Htz1 mutants that are not stably retained in chromatin[47,48] result in synergistic increases in GCR rates when combined with *mrc1Δ*, like *htz1Δ*. Thus, stable incorporation of Htz1 by SWR-C is required for preventing genome instability in the presence of *mrc1Δ*-induced replication damage.

An important clue to the defects caused by loss of SWR-C/Htz1 is the suppression of growth defects of the *mrc1Δ swr1Δ* double mutant strain by HR and the accumulation of aberrant DNA structures during replication in the *tof1Δ swr1Δ* and *tof1Δ htz1Δ* strains. These results argue that the replication damage occurring in the absence of Mrc1/Tof1/Csm3 and SWR-C/Htz1 is processed into substrates for HR, such as DSBs or single-stranded gaps, whereas HR plays a less important role in strains lacking Mrc1/Tof1/Csm3 when SWR-C/Htz1 is present. Potential roles for SWR-C/Htz1 that are consistent with the analysis of replication intermediates are: (1) normal incorporation of Htz1 into chromatin may help directly stabilize damaged replication forks by preventing fork reversal ("Fork Stability Model", Fig. 6); or (2) incorporation of Htz1 into chromatin could occur at sites of replication damage after the collapse of the replication fork and prevent processing of the replication damage to DSBs by helicases and endonucleases ("Collapsed Fork Stability Model", Fig. 6). This stabilization could occur via direct Htz1-replication fork interactions, which could slow the MCM DNA helicase when it becomes uncoupled from the remaining replisome, and/or by formation of a specialized chromatin domain around the damaged site that may occur after replication fork collapse, which may be consistent with the role of Htz1 in antagonizing the formation of Rad51 filaments at HO endonuclease-induced DSBs[25]. In either case, the formation of DSBs in the absence of Mrc1/Tof1/Csm3 and SWR-C/Htz1 is consistent with the requirement for HR in the *mrc1Δ swr1Δ* strain and the formation of aberrant DNA structures at replication origins.

SWR-C/Htz1 and Mrc1 are evolutionarily conserved. Their homologs are significantly mutated (SRCAP in prostate cancer and glioblastoma; CLASPIN in gliomas and breast cancer) or overexpressed (H2A.Z in liver, colorectal, and metastatic breast cancer) in human cancers, and these alterations are predicted to play important roles in carcinogenesis[66–71]. It will be interesting to examine whether the genetic interactions described here are conserved in mammalian cells and play a role in cancer development.

## Methods

**Strains and plasmids**. To test the effects of various mutations in the three GCR assays, the mutations were introduced into the strains RDKY7635 (dGCR), RDKY7964 (sGCR), and RDKY8625 (uGCR) using standard PCR-based gene replacement methods, and selected strains were also generated by genetic crosses between appropriate haploid parental strains. All gene deletions and mutations were verified by PCR amplification and/or Sanger sequencing. All strains were plated for single colonies and examined for the presence of more rapidly growing variants to ensure that the isolates used for individual experiments did not contain suppressors of growth defects. The genotypes of all strains are listed in Supplementary Table 6. The sequences of the primers used are listed in Supplementary Table 7.

The *mrc1-1-843* allele was constructed by replacing 759 bp at the C-terminus of *MRC1* with *kanMX4* amplified from plasmid pFA6a-*kanMX4*. The protein expressed from this allele contains amino acids 1-843 of Mrc1, followed by the 10-amino acid sequence RTLQVDGSPG; this C-terminal end differs from the 200-amino acid sequence added to the C-terminus of the protein encoded by the previously described *mrc1-C14* allele[30] (Supplementary Fig. 19); however, both alleles cause the same phenotypes including resistance to 200 mM HU, viability when combined with deletion of *RAD9*, and delayed progression through S phase (Supplementary Fig. 19)[30]. The *mrc1-aq* allele was introduced into *S. cerevisiae* prior to introduction of the *CAN1-URA3* cassette as follows: the *mrc1-aq* fragment from pET24/*mrc1-aq* (a gift from Huilin Zhou) was cloned into the plasmid pRS426 by gap repair, and *kanMX4* (amplified from pFA6a-*kanMX4*) was cloned into the *Eag*I restriction site downstream of *mrc1-aq* to obtain pRDK1779. The endogenous *MRC1* gene in RDKY7629 (for dGCR and sGCR assays) and RDKY8624 (for the uGCR assay) was first replaced with *URA3*, and subsequently, the *URA3* gene was replaced with the *mrc1-aq.kanMX4* fragment amplified from pRDK1779 to obtain RDKY8304 and RDKY8818, respectively. RDKY8304 was transformed with the *yel072w::CAN1-URA3* (amplified from plasmid pRDK1378 [49]) or the *yel068c::CAN1-URA3* cassette (amplified from pRDK1379 [49]) to generate the dGCR and sGCR *mrc1-aq* strains RDKY8305 and RDKY9089, respectively, and RDKY8818 was transformed with the *yel068c::CAN1-URA3* cassette (amplified from pRDK1379) to generate the uGCR *mrc1-aq* strain RDKY9091. The *htz1-K(126,133)R* allele was generated as follows: *HIS3* was amplified from pRS303 using a forward primer containing the C-terminal 91 nucleotides of *HTZ1* (nt 315–405) that included the K126R (c.377A>G) and K133R (c.398A>G) mutations. This amplicon was used to transform the desired strains, thereby introducing the two mutations in *HTZ1* and inserting *HIS3* downstream of the *HTZ1* stop codon. *HIS3* was also inserted similarly downstream of wild-type *HTZ1* as a control. To mutate Htz1 lysines 4, 9, 11, and 15 (referred to in previous studies as K3, K8, K10, and K14), the *HTZ1* ORF was cloned upstream of the *hph* locus in plasmid pFA6a-*hphNT1* using the *Hind*III and *Sal*I sites to obtain pRDK1834. The GeneArt Site-Directed Mutagenesis kit (Thermo Fisher Scientific) was used to simultaneously mutate *HTZ1* nucleotides A11, A26, A32, and A44 to G (for K→R mutations) or nucleotides A10, A25, A31, and A43 to C (for K→Q mutations) to obtain plasmids pRDK1835 and pRDK1836, respectively. The mutant *htz1* constructs including the *hphNT1* region were amplified and used to transform RDKY7635 and RDKY8301. Because the *hphNT1* marker caused an approximately threefold increase in the GCR rate in strains containing *mrc1::kanMX4*, the *hphNT1* marker was eventually replaced with the *HIS3* marker amplified from pRS303[72]; the presence of the *HIS3* marker downstream of wild-type *HTZ1* or mutant *htz1* alleles had no effect on the GCR rate. To introduce *EGFP* downstream of *DDC2* and *HUG1*, the relevant strains were transformed with an *eGFP-hphNT1* cassette amplified from plasmid pYM25[73] using gene-specific targeting primers. Strains used for 2-D gel electrophoresis experiments were constructed in the W303 strain background using standard PCR-based gene replacement methods, and their sensitivity to HU and MMS was verified.

**Determining GCR rates**. GCR rates were measured using fluctuation analysis by plating appropriate dilutions of saturated overnight cultures on YPD and GCR media[6,74]. For each strain, at least two independent biological isolates were tested using at least 14 independent cultures, and the median GCR rate and 95% confidence interval were calculated from the observed distributions of mutants[74]. *p* values for significance were calculated using the Mann-Whitney two-tailed test at the server http://vassarstats.net/utest.html.

**Whole-genome sequencing (WGS)**. Genomic DNA was prepared from *S. cerevisiae* strains using the Gentra/Puregene Yeast/Bact. kit (Qiagen). Libraries of 500–700 bp fragments were prepared from genomic DNA samples[74]. Briefly, genomic DNA samples were fragmented by sonication (Covaris) to obtain an average fragment size of 600 bp, and the fragments were blunt-ended and 5′-phosphorylated using the End-It DNA End Repair Kit (Epicentre Technologies). The DNA was then purified using the MinElute PCR purification kit (Qiagen), and

3′ ends were adenylated using Klenow DNA polymerase (NEB). Indexed Illumina adapters were then ligated to the A-tailed DNA fragments using Quick DNA Ligase (NEB), and the samples were purified using the MinElute kit. Size selection was performed using gel extraction to obtain 600–800 bp fragments, and the adapter-ligated fragments were enriched by PCR using the KAPA library amplification readymix (KAPA Biosystems) with primers AATGATACGGCGACCACCGA-GATCTACAC and CAAGCAGAAGACGGCATACGAGAT. The libraries were then purified using gel extraction to select for 600–800 bp fragments. The library concentrations were measured in a Qubit fluorometer using the Qubit dsDNA HS assay kit. Sets of 12 libraries (10 nM each) were mixed for multiplexing and sequenced on an Illumina Hi-Seq 2000 instrument using the Illumina GAII sequencing procedure for paired-end short-read sequencing to obtain 50-bp reads.

**Analysis of GCR structures from WGS data.** Individual reads from all read pairs were mapped to the S288c reference genome using bowtie. The Pyrus suite (https://sourceforge.net/projects/pyrus-seq/) was used to determine genomic alterations including GCR structures from mapped sequence reads[50].

**Drug sensitivity assay.** To test sensitivity to chronic exposure to HU and MMS, tenfold serial dilutions of cultures of selected strains grown in YPD medium at 30 °C were spotted on plates containing drugs at the indicated concentrations. Two independent isolates were tested for each strain, and the plating was performed in duplicate. The plates were photographed after 2 days of incubation at 30 °C or after 6 days for *rad52Δ*-containing strains. Representative images are shown.

**Measurement of doubling times.** Logarithmic-phase cultures grown in YPD medium at 30 °C were sampled at appropriate intervals, and their OD600 was measured using a Nanodrop spectrophotometer with 1.5-ml cuvettes. The doubling time was calculated from the logarithmic phase of the growth curves.

**Measurement of Ddc2-GFP foci.** Cells were grown in complete synthetic medium to log phase and examined by live imaging using an Olympus BX43 fluorescence microscope with a 60 × 1.42 PlanApo N Olympus oil-immersion objective. GFP fluorescence was detected using a Chroma FITC filter set and captured with a Qimaging QIClick CCD camera. Images were analyzed using Meta Morph Advanced 7.7 imaging software, keeping processing parameters constant within each experiment.

**Fluorescence-activated cell sorter (FACS) analysis.** Cell-cycle analysis was conducted using a standard protocol[75]. In brief, $1 \times 10^7$ cells were collected by centrifugation and resuspended in 70% ethanol for 16 h. Cells were then washed in 0.25 M Tris-HCl (pH 7.5), resuspended in the same buffer containing 2 mg/ml of RNaseA and incubated at 37 °C for at least 1 h, then treated overnight with pro-teinase K (1 mg per ml) at 37 °C. Cells were then resuspended in 200 mM Tris-HCl (pH 7.5) buffer containing 200 mM NaCl and 80 mM MgCl$_2$ and stained in the same buffer containing 1 μM Sytox-green (Invitrogen). Samples were then diluted tenfold in 50 mM Tris-HCl (pH 7.8) and analyzed using a Becton Dickinson FACScan instrument. This FACS analysis verified that all of the strains used in the experiments reported in this study were haploids.

To measure Hug1-GFP expression, 1-ml samples of logarithmic-phase cultures grown in YPD medium were centrifuged, and the cells were resuspended in 1 ml sterile water. The cells were sonicated using a Branson Digital Sonifier by applying five 1 s pulses at 10% amplitude, with a 1 s interval between pulses. The cells were then directly used for FACS analysis in a Becton Dickinson FACS instrument.

**Two dimensional (2D) gel electrophoresis.** Purification of DNA intermediates and 2D gel analysis were performed as previously described[76]. Approximately $2-4 \times 10^9$ cells (200-ml cultures) were arrested by addition of sodium azide to a final concentration of 0.1% and cooled on ice before psoralen crosslinking. Cells were washed, resuspended in 5 ml cold water, transferred to small petri dishes and placed on ice. Furthermore, 300 μl of 4,5′,8-trimethylpsoralen solution (0.2 mg per ml in 100% ethanol) was added prior to extensive resuspension by pipetting, followed by 5 min incubation in the dark and 10 min of UV irradiation at 365 nm (UV Stratalinker, Stratagene). The procedure was then repeated three times to ensure extensive crosslinking. Cells were harvested by centrifugation, washed in cold water, and incubated in 5 ml of spheroplasting buffer (1 M sorbitol, 100 mM EDTA (pH 8.0), 0.1% β-mercaptoethanol, and 50 U zymolyase/ml) for 1.5 h at 30 °C. Subsequently, 2 ml water, 200 μl RNase A (10 μg per ml), and 2.5 ml Solution I (2% w/v cetyl-trimethyl-ammonium-bromide (CTAB), 1.4 M NaCl, 100 mM Tris–HCl (pH 7.6), 25 mM EDTA (pH 8.0)) were sequentially added to the spheroplast pellets, and samples were incubated for 30 min at 50 °C. 200 μl Pro-teinase K (20 mg per ml) was then added, and the samples were incubated at 50 °C for 90 min and then shifted to 30 °C overnight. The samples were then centrifuged at 4000 rpm for 10 min. The cellular debris pellet was retained for further extraction, and the supernatant was extracted with 2.5 ml chloroform/iso-amylalcohol (24:1), and the DNA in the upper phase was precipitated by addition of 2 volumes of Solution II (1% w/v CTAB, 50 mM Tris–HCl (pH 7.6), 10 mM EDTA) followed by centrifugation at 8500 rpm for 10 min. The pellet was resuspended in 2 ml Solution III (1.4 M NaCl, 10 mM Tris–HCl (pH 7.6), 1 mM EDTA). Residual DNA in the cellular debris pellet was also extracted by resus-pension in 2 ml Solution III and incubation at 50 °C for 30 min, followed by extraction in 1 ml chloroform/isoamylalcohol (24:1). The upper phase was pooled with the main DNA prep. Total DNA was then precipitated with 1 volume of isopropanol, washed with 70% ethanol, air-dried, and finally resuspended in 1X TE. The genomic DNA samples were digested with *Eco*RV and *Hind*III, and signals were detected following 2D gel electrophoresis and standard Southern blot pro-cedures using a probe against *ARS305* (Chr III 39002–40063; indicated in Fig. 5a).

## Data availability

All relevant data is available from the authors upon reasonable request. All WGS reads are available at the National Center for Biotechnology Information Sequence Read archive under accession numbers SRP128125 and SRP128567.

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

## Acknowledgements

This work was supported by NIH grant R01-GM26017 to R.D.K., support from the Ludwig Institute for Cancer Research to R.D.K. and C.D.P., and by an Italian Association for Cancer Research (AIRC) grant IG 18976 and European Research Council Consolidator Grant 682190 to D.B.

## Author contributions

A.S., C.D.P. and R.D.K. conceived the overall experimental design. A.S. performed strain construction, quantitative rate measurements, drug sensitivity studies, and Hug1-GFP measurement. A.S. and C.D.P. analyzed the GCR structures derived from WGS data. B.-Z.L. analyzed Ddc2-GFP foci formation. B.S. and D.B. performed the 2-D gel electrophoresis and FACS experiments. A.S., C.D.P., D.B. and R.D.K. wrote the paper, and all other authors revised and modified the paper.

## Additional information

**Competing interests:** The authors declare no competing interests.

