## [Peer Review File · Nature Communications]

Reviewers' Comments:

Reviewer #1:

Remarks to the Author:

While the importance of genome stability is clear, the mechanisms that protect the genome remain poorly understood. In particular, the intersection between DNA replication, the replication checkpoint, and ultimately genomic instability is still unclear. To try to connect these processes, the authors here characterize the phenotype of a double mutant between a known replication and checkpoint protein Mrc1 and a histone variant Htz1 that has previously been associated with genomic instability. The authors show convincing data of an increase in gross chromosomal rearrangements (GCRs), replication defects, and sensitivity to replication stress.

Major comments

1. The interpretation of the *htz1 mrc1* double mutant phenotype is not sufficiently discussed. The authors show a genetic interaction between Htz1 and the replication function of Mrc1, and they interpret this to mean that Htz1 promotes the repair of damage caused by the absence of Mrc1. However, there is currently no evidence of a direct interaction of Htz1 and the replication fork where Mrc1 is acting, making the interpretation of the genetic interaction tricky since Mrc1 has genetic interactions with so many other proteins. Especially since Htz1 has been implicated in transcription as well as heterochromatin formation and chromosome segregation, Htz1 could easily indirectly affect replication, causing a synthetic phenotype with Mrc1. While experiments to show a direct interaction between Htz1 and Mrc1-dependent damage could be very difficult, there must be a significant discussion of alternative interpretations of the double mutant phenotype.
2. The question of endogenous replication stress checkpoint activation in the double mutants is not cleanly addressed. The two assays presented, Ddc2-foci and Hug1-GFP expression, show inconsistent results: the *mrc1-184* mutant phenocopies *mrc1* in the Ddc2-foci assay but not in the Hug1-GFP assay and the *swr1 mrc1* double mutant shows more activation in the Hug1-GFP assay but not in the Ddc2-foci assay. Further, *mrc1* mutants have previously been shown to induce checkpoint activation specifically in S phase (Alcasabas et al, 2001). Therefore, results using an asynchronous population are difficult to interpret; does an increase in the number of cells in the double mutant with Ddc2-GFP foci or high Hug1-GFP expression mean more cells are activating the checkpoint when in S phase, cells are activating the checkpoint outside of S phase, or more cells are in S phase and activate the checkpoint as in the single mutant? (Dhillon et al, 2006 showed cell cycle defects in *htz1* suggesting that the latter is certainly possible.) At least a synchronized experiment using Rad53 phosphorylation as a read-out would clarify the phenotype of the double mutants compared to the previously published phenotypes of the single mutants.

Minor comments

1. Please illustrate *mrc1*, *swr1 mrc1*, *htz1 mrc1*, and *swr1 htz1 mrc1* data from Table 1 graphically.
2. Why was there a sudden switch from the dGCR assay to uGCR assay within the Ino80 section? Are the results the same for both assays?
3. In the discussion, the statement "ino80 mutation causes additional defects" is made, but the data presented showed that *ino80 htz1 mrc1* cells had reduced GCRs compared to *htz1 mrc1* (and the conclusion in the results section was that Ino80 may be partially responsible for the increased GCR rates in the double mutants). Please clarify.
4. The finding that the *mrc1 1-843* mutant phenocopies *mrc1* provides strong support that the Mrc1 function in promoting efficient replication is what prevents GCRs. However, the data also shows that the *mrc1-aq* mutant has higher GCR in the uGCR assay but not the dGCR or sGCR assay. This would suggest that the checkpoint function of Mrc1 is important for reducing uGCRs - how does this fit in the overall model?
5. What are the error bars and n in Fig. 3? Particularly in Fig. 3D, are the error bars calculated on independent experiments or on the number of cells in the experiment? And, how was the Hug-GFP expression converted to fold change?
6. Why were the cells arrested for 4h in HU for Fig. S16? As shown in Fig. 5B, even wild-type cells

are asynchronously progressing through S phase after 4h of treatment, making the resulting cell-cycle progression analysis in Fig. S16 difficult to analyze. The authors should consider repeating the experiment with only a 2h treatment when the cells in wild-type as well as the mutants are more synchronously arrested in early S phase.

7. At the end of the section describing the cell cycle progression after HU treatment, the following statement is made, "Together these data suggest that loss of Mrc1 or Tof1-Csm3 in combination with loss of SWR-C/Htz1 leads to defects in replication and recovery from replication stress." The data specifically shows defects in replication during recovery from replication stress, not replication generally.

Reviewer #2:

Remarks to the Author:

The H2A.Z histone variant is localized to nucleosomes that flank promoters of genes transcribed by RNA polymerase II, as well as at the boundaries of heterochromatic regions, nucleosomes flanking replication origins, and near centromeres. In budding yeast, H2A.Z (and its deposition complex, SWR1C), play roles in gene expression, controlling the kinetics of transcriptional induction for many inducible genes. Although H2A.Z is mostly known for its roles in transcription in all eukaryotes, it is also known to play roles in genome stability pathways, where it stimulates recombinational repair of DNA double strand breaks (DSBs) by facilitating DSB resection. H2A.Z may also promote alternative DSB repair pathways by promoting the recruitment of recalcitrant DSBs to the nuclear periphery. Recent work has also implicated H2A.Z in DNA replication fidelity, and yeast that lack H2A.Z are somewhat sensitive to replication stress agents.

In a previous publication, the authors found that deletion of the SWR1 gene, which encodes the catalytic subunit of the H2A.Z deposition machinery, showed enhanced genome instability (as assayed by gross chromosomal rearrangements [GCRs]) when combined with an individual deletion of ~95 other yeast gene deletions (Nature Commun., 2016). Among this group of double mutants were *tof1 swr1* and *mrc1 swr1* doubles which showed enhanced GCR rates. In this current manuscript, the authors have extended this previous study with a comprehensive analysis of genome instability when a *swr1* or *htz1* (encoding H2A.Z) deletion is combined with either a *mrc1*, *tof1*, or *csm3* (not uncovered in the previous screen) deletion. The data largely confirms the previous analysis, showing that the single mutants have very little GCR phenotype, whereas the double mutants have extensive GCR phenotypes in several different quantitative assays. The authors perform an extensive sequencing analysis of GCR events in the wildtype and double mutant strains, but the authors find that the spectrum of GCR events is not altered much by loss of H2A.Z, other than the increased frequency. Furthermore, the role of H2A.Z does not seem to be linked to H2A.Z acetylation or sumoylation, so how it functions with this genome stability pathway remains unclear. In essence, most of the data simply shows that each single mutant leads to a weak amount of replication stress, whereas the double mutant leads to enhanced stress. This is basically consistent with multiple, redundant pathways for promoting replication fork integrity.

The authors imply that the double mutant interactions are somewhat unique between H2A.Z and the Mrc1/Tof1/Csm3 replication fork progression complex. However, the original screen identified negative genetic interactions of SWR1 with many other genes, including those that encode proteins that function in other genome instability pathways (e.g. Srs2, Ku70, Rev3, Rev7, Rad27, Rad17, Rad10, Mms2, Exo1, Elg1, Est1, Est3). Although *swr1* does not interact with all repair factors, there do appear to be many cases where crippling H2A.Z deposition makes many pathways function less efficiently.

The authors propose a model in which H2A.Z functions at the replication fork. However, there is no evidence for localized H2A.Z deposition at forks. Previous studies have only observed significant H2A.Z at stalled forks when the INO80C enzyme is inactivated, and in this case H2A.Z deposition was considered harmful for fork stability. What seems more likely is that it may be the pre-

existing, promoter-associated H2A.Z nucleosomes or H2A.Z nucleosomes that flank origins that facilitate fork stability in HU. In this model, H2A.Z nucleosomes may simply be a weaker barrier to fork progression. A more indirect model would propose that H2A.Z controls transcription of a key genome instability factor that is normally induced by stress. None of these models are ruled out in the current text.

In general, this is a solid genetics manuscript that provides a comprehensive analysis of interactions among H2A.Z and Mrc1/Tof1/Csm3, but there the work lacks mechanistic insights.

Specific points:

1. How were the strains constructed? For *ino80* mutants, it is well-known that these strains are extremely sick in most backgrounds (lethal in *w303*) and become polyploid very rapidly. Indeed, several groups have reported that a *swr1 ino80* double mutant is lethal. Much of the yeast community is now expecting tetrad dissection for making double mutants, especially if the double mutants are quite sick, as second site suppressors can arise very rapidly.

2. Page 6. The authors cite the recent work from Jentsch and colleagues showing that INO80 requires H2A.Z for recruitment to DSBs. However, this has not to my knowledge been shown for replication forks. Furthermore, loss of H2A.Z deposition actually suppresses the need for INO80 at DSBs, as does constitutive acetylation of H2A.Z during replication stress. These data have led to the model that the role of INO80 is to antagonize H2A.Z at DSBs and at forks, likely by catalyzing its removal and replacement from chromatin. This is quite different from the model discussed by the authors.

3. Page 7, first line. The term "SWR-C/Htz1 defects" is not really an accurate statement, as defects refers to the phenotype of mutants. This should be changed.

Reviewer #3:

Remarks to the Author:

The authors have in this manuscript investigated the effect of deletion of SWR1 or HTZ1, MRC1, CSM3 or TOF1 on gross chromosomal rearrangements (GCR) rates. Single deletions gave essentially no effect (over-lapping confidence intervals), whereas double mutants had severe replication defects that, according to the authors, give an increased rate of GCR's. The genetic studies are complemented by 2D-gel electrophoresis to monitor the fork progression and possible intermediates. In addition are selected yeast isolates with GCR's whole genome sequenced to confirm where the GCR's are located. At the end, the authors propose a model in which the presence of Htz1 prevents aberrant replication intermediates that occurs in *mrc1*, *tof1*, or *csm3* mutants to be converted to double-stranded DNA breaks.

Major comments

1. The manuscript should be reorganized. It is very difficult to follow the rationale and comprehend the results. The authors have made a very comprehensive analysis, combining many different genes in three separate genetic assays. The authors are jumping between the genetic assays with different readouts and at the same time are different effects of modified genes discussed in detail. Please revise and improve the clarity.

2. All tables should be revised. Please present all values of the rates in a column with the same exponent. For example, Table 1 dGCR assay (rate) the exponent varies between 10×10^{-8} to 10×10^{-6} , but should be kept at 10×10^{-8} throughout the column to allow the reader to easily apprehend if the confidence intervals are overlapping or not. This should be corrected for in all tables, including supplemental tables. I have not gone through all the data since I must rewrite the tables to be able to read them, but a large portion of the reported data show very large and

overlapping confidence intervals when adjusted to show the same exponent.

3. My largest concern with this manuscript is how the yeast strains were constructed. All yeast strains were propagated as haploids and were constructed by sequential deletion of genes that are known to confer genomic instability. This is a very dangerous approach since suppressor mutations may appear and they will be selected for. The variation in colony size in figure 1C (mrc1del column) corroborates my concerns. Under these circumstances should strains be maintained and constructed as heterozygote diploids. Only just prior to when experiments are performed should the haploid strains be dissected out to minimize the number of cell divisions. How can we trust that the reported rates are not affected by suppressor mutations?

4. On what numbers were the rates and confidence intervals based? How many isolates from each strain were analyzed? Please add to all tables with rates.

In conclusion, based on the design of the experiments I do not feel confident in the model which is proposed at the end of the paper.

Response to Reviewer's comments

General

I would like to thank the reviewers for their comments, which were helpful in revising the paper. In this section I summarize some of the overall changes and modifications to the paper to address some of the general points made by the reviewers and in the following sections I provide each reviewer comment in italics and then address each specific point made by the reviewers in the plain text that follows the italics.

1. We have repeated a number of experiments including making new strains and added some additional checkpoint analysis, which I believe addresses key points made by the reviewers. As a consequence, we have expanded and modified the paper (P9-P10).
2. We have included some analysis of published gene expression data in strains defective for SWR-C/Htz1 that we feel excludes the possibility that the defects in SWR-C/Htz1 are indirectly affecting GCRs through altered expression of GCR-suppressing genes and replication genes. We also comment on issues of chromatin structure. This analysis is inconsistent with an indirect role for SWR-C/Htz1 defects being due to an indirect effect (P5, L29 to P6, L9 and P16, L4-10).
3. We have added additional information about methods to clarify points raised by the reviewers or to add details that required additional explanation based on feedback from the reviewers (Supplementary Data 1; P17, L20-26; P19, L4-8; P20, L18-19; Fig 3 legend; Table 1 legend).
4. Two of the reviewers have questioned our strain construction methods and at least one reviewer has speculated that all of our results could be problematic because our strain construction methods. I disagree with this. But to address this issue, we have added additional information about our strain construction methods and strains to the paper as well as well as adding additional data.

First, there is nothing sacrosanct about constructing strains by crosses. It takes exactly the same number of cell divisions to go from a spore to the cultures needed for analysis as it does for a transformed cell so the possibility of suppressors exists for both methods. In addition, the double strand break repair that occurs on each chromosome during meiosis is associated with lower fidelity DNA synthesis providing additional opportunity for genetic variation in strains constructed by crosses.

Second, it is careful analysis of the strains during growth and not tetrad analysis that allows the geneticist to detect growth suppressors. We perform this type of analysis with all strains and have now stated this in the methods (P17, L20-26). We did not observe any suppressors in our strains. Indeed, most of the strains studied here were constructed by crosses in our original study and then validated here by reconstructing them using a different method, which is the proper way to do yeast genetics; the results obtained in these two studies were consistent with each other.

Third, defects in the genes encoding the Swr1 complex and Htz1 cause little or no growth defect in combination with either *tof1* or *csm3* mutations. Defects in the genes encoding the Swr1 complex and Htz1 show some growth defects in combination with *mrc1* mutations. However, we carefully examine all strains for signs of accumulating

growth suppressors by observing their growth on non-selective media and we have never seen growth suppressors arise in the double mutants. In addition, a reason why we examine independent isolates of each strain in all of our experiments, as well as test all genes in complexes, is to allow ample opportunity to detect growth suppressors. This information has been added to the methods section (P17, L20-26).

Fourth, I point out that our whole genome sequence analysis presented in the supplement provides enormous detail about strain ploidy. We observed in the vast majority of the double mutants where we selected for a GCR, the only altered chromosome was the GCR chromosome and all other chromosomes were at their normal complement and had their normal sequence. This has been pointed out in the results (P8, L19-21) and the fact that we have also performed extensive ploidy analysis of our strains by FACs is now included in the methods (P20, L18-19). We did not observe significant issues with polyploidy.

Finally, we constructed selected single and double mutants by crosses and determined GCR rates. We found no difference between the strains constructed by crosses with those constructed by transformation. The new rates have been included in the legend to Table 1 for comparative purposes.

5. We have also modified and expanded the hypotheses as to how defects in Mrc1/Tof1/Csm3 and SWR-C/Htz1 lead to increased GCRs (P16 bottom – P17 top; Figure 6).

6. We have many a large number of changes throughout the paper in order to provide a clear description of the studies performed. Those changes that are specific to individual reviewer comments are cross-referenced to the specific comments.

Reviewer 1

While the importance of genome stability is clear, the mechanisms that protect the genome remain poorly understood. In particular, the intersection between DNA replication, the replication checkpoint, and ultimately genomic instability is still unclear. To try to connect these processes, the authors here characterize the phenotype of a double mutant between a known replication and checkpoint protein Mrc1 and a histone variant Htz1 that has previously been associated with genomic instability. The authors show convincing data of an increase in gross chromosomal rearrangements (GCRs), replication defects, and sensitivity to replication stress.

We thank the reviewer for what seems like a positive view of our paper. I hope that our modifications have addressed their concerns.

Major comments

1. The interpretation of the htz1 mrc1 double mutant phenotype is not sufficiently discussed. The authors show a genetic interaction between Htz1 and the replication function of Mrc1, and they interpret this to mean that Htz1 promotes the repair of damage caused by the absence of Mrc1. However, there is currently no evidence of a direct interaction of Htz1 and the replication fork where Mrc1 is acting, making the interpretation of the genetic interaction tricky since Mrc1 has genetic interactions with so

many other proteins. Especially since Htz1 has been implicated in transcription as well as heterochromatin formation and chromosome segregation, Htz1 could easily indirectly affect replication, causing a synthetic phenotype with Mrc1. While experiments to show a direct interaction between Htz1 and Mrc1-dependent damage could be very difficult, there must be a significant discussion of alternative interpretations of the double mutant phenotype.

These are good points. First, we have included analysis of published gene expression data in strains defective for SWR-C/Htz1 that we feel excludes the possibility that defects in SWR-C/Htz1 have indirect effects on GCR rates through altered expression of GCR-suppressing genes and replication genes (P5, L29 to P6, L9 and P16, L4-10). We also comment on issues of chromatin structure. This analysis is inconsistent with an indirect role for SWR-C/Htz1. Second, we have revised and expanded our discussion of different models. We were not trying to suggest that there was a direct interaction between Htz1 and replication forks. While our 2D gel and FACS analysis indicates that at least under conditions of replication stress, SWR-C/Htz1 facilitates accurate fork processing and replication completion, we thought it most likely that the effect observed was related to the broader incorporation of Htz1 into chromatin (See P14, L30 and P16, L26, 27). I hope this is clearer now.

2. The question of endogenous replication stress checkpoint activation in the double mutants is not clearly addressed. The two assays presented, Ddc2-foci and Hug1-GFP expression, show inconsistent results: the mrc1-1-843 mutant phenocopies mrc1 in the Ddc2-foci assay but not in the Hug1-GFP assay and the swr1 mrc1 double mutant shows more activation in the Hug1-GFP assay but not in the Ddc2-foci assay.

We repeated these experiments with reconstructed strains and I think the reviewer will now see that the inconsistency has been eliminated and that the mrc1-1-843 mutation behaves the same as the mrc1 Δ mutation (See Figures 3 and S14, and P10, L4-15).

Further, mrc1 mutants have previously been shown to induce checkpoint activation specifically in S phase (Alcasabas et al, 2001). Therefore, results using an asynchronous population are difficult to interpret; does an increase in the number of cells in the double mutant with Ddc2-GFP foci or high Hug1-GFP expression mean more cells are activating the checkpoint when in S phase, cells are activating the checkpoint outside of S phase, or more cells are in S phase and activate the checkpoint as in the single mutant? (Dhillon et al, 2006 showed cell cycle defects in htz1 suggesting that the latter is certainly possible.) At least a synchronized experiment using Rad53 phosphorylation as a read-out would clarify the phenotype of the double mutants compared to the previously published phenotypes of the single mutants.

We agree that this is a good point. While the suggested Rad53 phosphorylation experiment is certainly reasonable, we addressed this question more directly by doing a morphology analysis of the cultures used for the Ddc2-GFP foci assay experiments. A simple summary of what we found is that the mutants with an increased frequency of cells with Ddc2-GFP foci had a greater fraction of cells with Ddc2-GFP foci among the small/mid budded cells (S-phase cells). This supports the view that we are observing increased checkpoint activation and not just an increased proportion of S-phase cells. These data are included in the results (P10, L15-24).

Minor comments

1. Please illustrate *mrc1*, *swr1 mrc1*, *htz1 mrc1*, and *swr1 htz1 mrc1* data from Table 1 graphically.

We respectfully decline to do this. This request and the request from reviewer 3 are mutually exclusive so we decided to go with reformatting the tables as requested by reviewer 3, which I think makes the results much clearer. I have used a graphical illustration in some cases but find that this approach isn't as useful as a table. In cases like our data, the large range of rates covering an approximately 200-fold range of increases results in a graphical illustration making it very difficult to see what the exact GCR rates are without a now duplicative rate table. This is in contrast to the example of Van et al 2015 (Ref 25) where the effects are spread over only an 8-fold range and where a graphical illustration does not obscure what the exact rates are.

2. *Why was there a sudden switch from the dGCR assay to uGCR assay within the Ino80 section? Are the results the same for both assays?*

In response to Reviewer 2 specific point 1, we have deleted the section on Ino80. This was a minor part of our analysis and never entered into our mechanistic hypotheses. See below for more information.

3. *In the discussion, the statement "ino80 mutation causes additional defects" is made, but the data presented showed that ino80 htz1 mrc1 cells had reduced GCRs compared to htz1 mrc1 (and the conclusion in the results section was that Ino80 may be partially responsible for the increased GCR rates in the double mutants). Please clarify.*

In response to Reviewer 2 specific point 1, we have deleted the section on Ino80. This was a minor part of our analysis and never entered into our mechanistic hypotheses. See below for more information.

4. *The finding that the mrc1-1-843 mutant phenocopies mrc1 provides strong support that the Mrc1 function in promoting efficient replication is what prevents GCRs. However, the data also shows that the mrc1-aq mutant has higher GCR in the uGCR assay but not the dGCR or sGCR assay. This would suggest that the checkpoint function of Mrc1 is important for reducing uGCRs - how does this fit in the overall model?*

We point out that even in the uGCR case, the effect of the *mrc1-aq* mutation is much smaller than that of either the *mrc1* deletion mutation or the *mrc1-1-843* mutation and have noted this in the text (P7, L9-12). We also note that the uGCR assay is highly selective for some types of GCRs, most notably de novo telomere additions, which may be produced by slower mechanisms (for example, telomere synthesis at 1 or 2 TGs) that are known to be physically coupled to checkpoint functions, and this could account for the checkpoint effect in the uGCR assay. Given that this is a small effect seen in only one assay that was not studied further, I would prefer to not speculate about this.

5. *What are the error bars and n in Fig. 3? Particularly in Fig. 3D, are the error bars calculated on independent experiments or on the number of cells in the experiment? And, how was the Hug-GFP expression converted to fold change?*

These points have all been clarified in the legend to Figure 3 on P23.

6. Why were the cells arrested for 4h in HU for Fig. S16? As shown in Fig. 5B, even wild-type cells are asynchronously progressing through S phase after 4h of treatment, making the resulting cell-cycle progression analysis in Fig. S16 difficult to analyze. The authors should consider repeating the experiment with only a 2h treatment when the cells in wild-type as well as the mutants are more synchronously arrested in early S phase.

The FACS experiments in Figure S16 were performed in order to address if the mutants have problems in replicating in the presence of HU (G1 synchronized cells released in the presence of HU for 4 hours) and subsequently, in recovering from replication stress. HU 4 hr was chosen as experimental condition because it allows WT cells to reach mid-S phase and therefore potential problems in replication fork progression in the mutants can be more accurately judged. Using HU 2 hr when most mutants appear to be early in S phase, would obscure this point.

We note that we performed 2D gel and FACS experiments when cells were released in the presence of HU; analysis of replication intermediates revealed that the replication problems are very severe in the double mutants already at HU 2hr (please refer to text and Figure 5). Therefore, it is likely that similar recovery defects are to be observed when cells are released from after 2 hr of HU treatment.

7. At the end of the section describing the cell cycle progression after HU treatment, the following statement is made, "Together these data suggest that loss of Mrc1 or Tof1-Csm3 in combination with loss of SWR-C/Htz1 leads to defects in replication and recovery from replication stress." The data specifically shows defects in replication during recovery from replication stress, not replication generally.

We have revised this statement.

Reviewer 2

a. The H2A.Z histone variant is localized to nucleosomes that flank promoters of genes transcribed by RNA polymerase II, as well as at the boundaries of heterochromatic regions, nucleosomes flanking replication origins, and near centromeres. In budding yeast, H2A.Z (and its deposition complex, SWR1C), play roles in gene expression, controlling the kinetics of transcriptional induction for many inducible genes.

I think the reviewer is suggesting these transcriptional changes could result in indirect effects that could account for the increased genome instability we see. We have included analysis of published gene expression data in strains defective for SWR-C/Htz1 that we feel excludes the possibility that defects in SWR-C/Htz1 have indirect effects on GCR rates through altered expression of GCR-suppressing genes and replication genes. We also comment on issues of chromatin structure. This analysis is inconsistent with the effects of SWR-C/Htz1 defects being due to an indirect effect. See P5, L29 to P6, L9 and P16, L4-10.

b. Although H2A.Z is mostly known for its roles in transcription in all eukaryotes, it is also known to play roles in genome stability pathways, where it stimulates recombinational repair of DNA double strand breaks (DSBs) by facilitating DSB resection. H2A.Z may also promote alternative DSB repair pathways by promoting the recruitment of recalcitrant DSBs to the nuclear periphery. Recent work has also implicated H2A.Z in

DNA replication fidelity, and yeast that lack H2A.Z are somewhat sensitive to replication stress agents.

All of these issues were discussed in both the introduction and discussion of our paper. We have now expanded and clarified these points throughout the paper. I do note that the effects the reviewer refers to from published studies are very small compared to the effects we see here. For example, the paper the reviewer is likely referring to in regard to replication stress is Van et al 2015 (Ref 25), which looked at loss of SWR-C/Htz1 in combination with a DNA polymerase mutation that results in modestly increased base misincorporation resulting in a 2-fold increase the rate of accumulating point mutations in the double mutants tested compared to the much larger effects seen here. Further, Chen et al 2012 Nature 489:576-80 who looked at chromatin remodeling factors in DSB resection excluded Swr-C as playing a role in resection. These defects (point mutation accumulation and long range resection) are fundamentally different than the correlated replication defects and genome instability analyzed in our study. Moreover, the magnitude of the effects in previous studies are substantially smaller than the much larger increases in GCR rates documented in our study. Moreover, we have directly tested *htz1* mutations that disrupt the role of Htz1 in both cohesion and recruitment of recalcitrant DSBs to the nuclear periphery and found that these processes did not explain the role of Htz1 in suppressing the formation of GCRs in strains with defects in *Mrc1/Tof1/Csm3*. I hope the reviewer agrees that we did discuss these issues and that our results considerably extend these prior observations.

*c. In a previous publication, the authors found that deletion of the SWR1 gene, which encodes the catalytic subunit of the H2A.Z deposition machinery, showed enhanced genome instability (as assayed by gross chromosomal rearrangements [GCRs]) when combined with an individual deletion of ~95 other yeast gene deletions (Nature Commun., 2016). Among this group of double mutants were *tof1 swr1* and *mrc1 swr1* doubles which showed enhanced GCR rates. In this current manuscript, the authors have extended this previous study with a comprehensive analysis of genome instability when a *swr1* or *htz1* (encoding H2A.Z) deletion is combined with either a *mrc1*, *tof1*, or *csm3* (not uncovered in the previous screen) deletion. The data largely confirms the previous analysis, showing that the single mutants have very little GCR phenotype, whereas the double mutants have extensive GCR phenotypes in several different quantitative assays.*

I do not agree with the flavor of this comment. Indeed, we did previously publish an extensive genetic screen, using semi-quantitative scoring methods that detected many genetic interactions. The interactions studied here were the most robust seen for SWR-C/Htz1 in the screen (See P5, L3-10) and are surely worthy of extensive validation and extensive expanded study as we have done here. It is inaccurate to suggest that we have just confirmed past results and further to suggest that the results of genetic screens are not worthy of either validation or extension by the vastly expanded study as we have done here seems unscientific and not what the field expects.

d. The authors perform an extensive sequencing analysis of GCR events in the wildtype and double mutant strains, but the authors find that the spectrum of GCR events is not altered much by loss of H2A.Z, other than the increased frequency. Furthermore, the role of H2A.Z does not seem to be linked to H2A.Z acetylation or sumoylation, so how it functions with this genome stability pathway remains unclear. In essence, most of the data simply shows that each single mutant leads to a weak amount of replication stress,

whereas the double mutant leads to enhanced stress. This is basically consistent with multiple, redundant pathways for promoting replication fork integrity.

The reviewer's comment suggests that our explanation of the structural analysis of the GCRs was not clear. Not only were the rates of accumulating GCRs increased but there were changes in the GCR spectrum. We have therefore provided an expanded description of the analysis performed and a clearer description of the results (P7, L27 to P8, L10 and P8, L11 to P9, L25). I further note that the type of structural analysis presented here defines the state of the art for the field that I hope future studies will emulate.

This comment in regards to replication stress and redundancy fails to acknowledge the massive synergy we observed in our studies or the novelty of this observation compared to previous reports in this area. See response to comment b above in regard to the magnitude of the genetic interactions seen.

e. The authors imply that the double mutant interactions are somewhat unique between H2A.Z and the Mrc1/Tof1/Csm3 replication fork progression complex. However, the original screen identified negative genetic interactions of SWR1 with many other genes, including those that encode proteins that function in other genome instability pathways (e.g. Srs2, Ku70, Rev3, Rev7, Rad27, Rad17, Rad10, Mms2, Exo1, Elg1, Est1, Est3). Although swr1 does not interact with all repair factors, there do appear to be many cases where crippling H2A.Z deposition makes many pathways function less efficiently.

We did not mean to imply that the genetic interactions between mutations affecting SWR-C/Htz1 and Mrc1/Tof1/Csm3 were the only interactions involving mutations affecting SWR-C/Htz1. Rather, these interactions were the strongest and most reproducible involving all of the relevant genes tested compared to other genetic interactions. We therefore focused our study on the most robust, strongest interactions, which were those affecting SWR-C/Htz1 and Mrc1/Tof1/Csm3. Of course, we hope to investigate other genetic interactions in future studies. We have expanded our explanation of this in the paper (P5, L3-10).

f. The authors propose a model in which H2A.Z functions at the replication fork. However, there is no evidence for localized H2A.Z deposition at forks. Previous studies have only observed significant H2A.Z at stalled forks when the INO80C enzyme is inactivated, and in this case H2A.Z deposition was considered harmful for fork stability. What seems more likely is that it may be the pre-existing, promoter-associated H2A.Z nucleosomes or H2A.Z nucleosomes that flank origins that facilitate fork stability in HU. In this model, H2A.Z nucleosomes may simply be a weaker barrier to fork progression. A more indirect model would propose that H2A.Z controls transcription of a key genome instability factor that is normally induced by stress. None of these models are ruled out in the current text.

I think our analysis of the effects on loss of Htz1 on transcription excludes the reviewer's indirect effect model. See response to comment b above and general comment 2. We have expanded our discussion of models and have hopefully made it clearer that we think that pre-existing Htz1 nucleosomes and their loss could also account for the effects seen. We never intended to suggest that there must be localized Htz1 deposition at replication forks and hope that is clear now. See P14, L30, P16, L26, 27 and the legend to Figure 6.

In general, this is a solid genetics manuscript that provides a comprehensive analysis of interactions among H2A.Z and Mrc1/Tof1/Csm3, but there the work lacks mechanistic insights.

I disagree with this comment. While our paper contains numerous genetics experiments, many of the genetic experiments involve separation-of-function alleles that cause known functional defects. Moreover, we also perform a number of mechanistic experiments including those on checkpoint activation, analysis of GCR structures and analysis of replication intermediates. The results from these experiments allow us to propose mechanistic hypotheses about the origin of genome instability in the various mutants studied. In my view, our studies are highly mechanistic and go far beyond what would result from a solely genetic study. I do appreciate the fact that the reviewer considers our work solid.

Specific points:

1. How were the strains constructed? For ino80 mutants, it is well-known that these strains are extremely sick in most backgrounds (lethal in w303) and become polyploid very rapidly. Indeed, several groups have reported that a swr1 ino80 double mutant is lethal. Much of the yeast community is now expecting tetrad dissection for making double mutants, especially if the double mutants are quite sick, as second site suppressors can arise very rapidly.

We have addressed the general point on strain construction under general responses point 2 and added additional information to the results, methods and tables where appropriate. We have carefully evaluated our strains for the presence of growth suppressors and polyploidy and have not detected any. Further, we show that GCR rates determined with strains constructed by transformation are the same as those determined with strains constructed with crosses. I hope this resolves the issue and allays the reviewer's concerns.

With regard to the points raised on ino80 mutants, the reviewer is basically suggesting that the literature on Ino80 and Swr1 is controversial with no agreement on whether ino80 and swr1 mutations are synthetically lethal, and these studies are further complicated by whether the strains become polyploid. We cannot resolve this field-wide issue here. We only did a limited experiment with an ino80 mutation because such studies are in the literature, but Ino80 was a minor topic of our paper. We have simply deleted the one experiment we did with ino80 mutations and do not feel this has altered our conclusions, especially given the reviewer's concerns about Ino80 in general.

2. Page 6. The authors cite the recent work from Jentsch and colleagues showing that INO80 requires H2A.Z for recruitment to DSBs. However, this has not to my knowledge been shown for replication forks. Furthermore, loss of H2A.Z deposition actually suppresses the need for INO80 at DSBs, as does constitutive acetylation of H2A.Z during replication stress. These data have led to the model that the role of INO80 is to antagonize H2A.Z at DSBs and at forks, likely by catalyzing its removal and replacement from chromatin. This is quite different from the model discussed by the authors.

As discussed under point 2 above, we have deleted the one experiment on Ino80. I note that our models focus on Htz1 more generally in chromatin and not on recruitment of

Htz1 to replication forks.

3. Page 7, first line. The term “SWR-C/Htz1 defects” is not really an accurate statement, as defects refers to the phenotype of mutants. This should be changed.

As requested, we have corrected this nomenclatural issue throughout the paper.

Reviewer 3

The authors have in this manuscript investigated the effect of deletion of SWR1 or HTZ1, MRC1, CSM3 or TOF1 on gross chromosomal rearrangements (GCR) rates. Single deletions gave essentially no effect (over-lapping confidence intervals), whereas double mutants had severe replication defects that, according to the authors, give an increased rate of GCR's. The genetic studies are complemented by 2D-gel electrophoresis to monitor the fork progression and possible intermediates. In addition are selected yeast isolates with GCR's whole genome sequenced to confirm where the GCR's are located. At the end, the authors propose a model in which the presence of Htz1 prevents aberrant replication intermediates that occurs in mrc1, tof1, or csm3 mutants to be converted to double-stranded DNA breaks.

I am pleased that the reviewer recognizes the extensive experimentation that has gone into this study. I hope our modifications have addressed the reviewer's concerns listed below.

Major comments

1. The manuscript should be reorganized. It is very difficult to follow the rationale and comprehend the results. The authors have made a very comprehensive analysis, combining many different genes in three separate genetic assays. The authors are jumping between the genetic assays with different readouts and at the same time are different effects of modified genes discussed in detail. Please revise and improve the clarity.

I hope that our extensive modifications to the paper have addressed this concern.

2. All tables should be revised. Please present all values of the rates in a column with the same exponent. For example, Table 1 dGCR assay (rate) the exponent varies between 10×10^{-8} to 10×10^{-6} , but should be kept at 10×10^{-8} throughout the column to allow the reader to easily apprehend if the confidence intervals are overlapping or not. This should be corrected for in all tables, including supplemental tables. I have not gone through all the data since I must rewrite the tables to be able to read them, but a large portion of the reported data show very large and overlapping confidence intervals when adjusted to show the same exponent.

We have reformatted the tables as requested. This and the presence of the fold-changes should simplify comparisons. The reviewer is mistaken about their view that the 95% confidence intervals are largely overlapping for the key comparisons made in this study. In none of the key comparisons between single and key double mutants are the 95% confidence intervals overlapping. To further facilitate evaluation of significance, we have extensively determined p-values for the significance of differences between

different GCR rates using Mann Whitney 2-tailed tests and provide this data in an extensive Supplementary Data Set. The method for this is indicated in the methods section (P19, L6-9). I am confident that if the reviewer examines all of the 95% confidence intervals and the p values, they will agree that all of the comparisons we discuss as well as others are significant.

3. My largest concern with this manuscript is how the yeast strains were constructed. All yeast strains were propagated as haploids and were constructed by sequential deletion of genes that are known to confer genomic instability. This is a very dangerous approach since suppressor mutations may appear and they will be selected for. The variation in colony size in figure 1C (mrc1del column) corroborates my concerns. Under these circumstances should strains be maintained and constructed as heterozygote diploids. Only just prior to when experiments are performed should the haploid strains be dissected out to minimize the number of cell divisions. How can we trust that the reported rates are not affected by suppressor mutations?

We have addressed this point under general responses point 2 and added additional information to the results, methods and tables where appropriate. We have carefully evaluated our strains for the presence of growth suppressors and polyploidy and have not detected any. Further, we show that GCR rates determined with strains constructed by transformation are the same as those determined with strains constructed with crosses. See P5, L25-28, P17, L20-26 and Table 1.

The reviewer suggests that the experiment in figure 1C documents the presence of growth suppressors. The reviewer appears to misunderstand this experiment or possibly the implications of selecting for GCRs in the patch test. In this experiment, patches of mutants were replicated onto toxic media that selects for the presence of GCRs. Each colony that grows up in the patch seen on the plate in the figure does so because it results from a single cell that contains a GCR. These GCRs both delete genes and duplicate other genes, and each GCR is different. It is the presence of the GCR that alters the growth rate and hence colony size due to the fact that each colony is now genetically different from the other colonies that grow up in the patch due to its own unique GCR. The colony size distribution reflects the GCRs and not the presence of suppressors in the starting strain. I further add that growth suppressors in sick strains result in much larger colony size differences than seen on these plates.

4. On what numbers were the rates and confidence intervals based? How many isolates from each strain were analyzed? Please add to all tables with rates.

We have added all of this information to the methods section (P19, L4-8). In all cases we used a minimum of 2 independent isolates of each strain and a minimum of 14 independent cultures. Also note that we added an extensive statistical analysis in Supplementary Data 1.

In conclusion, based on the design of the experiments I do not feel confident in the model which is proposed at the end of the paper.

I hope we have allayed the reviewer's concerns through additional information, re-writing and additional experimental data.

Reviewers' Comments:

Reviewer #1:

Remarks to the Author:

The authors have made substantial changes to the manuscript with marked improvement. However, there are still some remaining issues to be addressed.

1. The authors have provided analysis of transcriptional data in the *htz1Δ* mutant and find that there is little change in the mutant. This finding, however, does not eliminate the possibility of transcriptional changes in the *mrc1Δhtz1Δ* double mutant, which would be an indirect effect on the double mutant phenotype. This should be explicitly mentioned.
2. Determining Ddc2-foci specifically in small-budded cells does help to determine the amount of constitutive checkpoint activation in the mutants. However, the authors only provide data for the single *mrc1Δ* and *htz1Δ* mutants, not the double mutant. The *mrc1Δ* mutant is already known to have constitutive replication stress, and the *htz1Δ* mutant is already known to not have constitutive replication stress. The interesting question is about the double mutant phenotype.
3. It is still unclear in Fig 3D (and Fig 3E and Fig 3F) how the mean and standard error were calculated. What is the n of these experiments – individual experiments or individual cells from one (or multiple) experiments?
4. The authors argue that the FACS experiments in Fig S16 were done to address both the question of replicating in the presence of HU and after removal of replication stress. The authors have shown that the mutants accumulate aberrant replication structures even early in S phase in the presence of HU and reach a different overall replication state after 4h (Fig 5B). Therefore, the recovery from HU arrest cannot be cleanly addressed with the mutants since they start from different states.

Reviewer #2:

Remarks to the Author:

In the revised manuscript, the authors have addressed many of my previous concerns. I think we will need to agree to disagree on the value of tetrad dissection compared to transformations when it comes to dealing with suppressor analyses. It does no good to argue in this forum. I think the key point is that several of the key strains have apparently been made by dissection and that several strains have also been shown to be free of second site suppressors by sequence analyses.

I think the authors have not quite understood my comments with regards to an indirect, transcriptional role for Htz1. In general Htz1 only impacts induced transcription, not expression of constitutively expressed genes. Consequently, I believe (though I may be incorrect) that the data sets evaluated by the authors did not include RNA isolated from DNA damage induced cells or cells undergoing replication stress. If Htz1 is key for stress-induced gene expression, then this may be key for the phenotypes observed.

In general this is a solid study.

Reviewer #3:

Remarks to the Author:

First, I want to emphasize that I enjoyed reading this new version of the manuscript. It was much improved, including the tables. The authors have clarified how the strains were constructed and this allows each reader to make their own evaluation of the results. I do appreciate that the authors clarify how they minimize the risk for suppressors of growth defects.

Response to Editor's and Reviewer's comments.

I would like to thank the reviewers for their additional helpful comments. We are pleased that they have so markedly improved their opinion of our study. We hope that the additional changes and comments indicated below will now lead to our paper being accepted for publication.

General

We have done some shortening of the manuscript including eliminating a number of references. We have also reformatted the manuscript, figures and tables to meet the requirements of the journal.

Reviewer #1 (Remarks to the Author):

The authors have made substantial changes to the manuscript with marked improvement. However, there are still some remaining issues to be addressed.

1. The authors have provided analysis of transcriptional data in the htz1Δ mutant and find that there is little change in the mutant. This finding, however, does not eliminate the possibility of transcriptional changes in the mrc1Δhtz1Δ double mutant, which would be an indirect effect on the double mutant phenotype. This should be explicitly mentioned.

We have added a note in this regard to the Results and Discussion on P6, L2-6 and P16, L6-9. Also see Reviewer 2, comment 2 for more discussion related to this point.

2. Determining Ddc2-foci specifically in small-budded cells does help to determine the amount of constitutive checkpoint activation in the mutants. However, the authors only provide data for the single mrc1Δ and htz1Δ mutants, not the double mutant. The mrc1Δ mutant is already known to have constitutive replication stress, and the htz1Δ mutant is already known to not have constitutive replication stress. The interesting question is about the double mutant phenotype.

We have done additional experimentation in regard to this point and have now added the relevant data for the mrc1Δ swr1Δ and mrc1Δ htz1Δ double mutants. What the results show is that the double mutants have a slightly higher frequency of Ddc2 foci in the small/medium budded cells and no major increase in the frequency of small/medium budded cells. This indicates that the induction of checkpoint activation is due almost entirely to the mrc1Δ mutation. These results confirm the results and conclusions reached through study of both Ddc2 foci and Hug1 expression in bulk cells. The data have been added to the text on P10, L10-23.

3. It is still unclear in Fig 3D (and Fig 3E and Fig 3F) how the mean and standard error were calculated. What is the n of these experiments – individual experiments or individual cells from one (or multiple) experiments?

I thought we had addressed this. However, we have now added additional explanation to the legend to Figure 3.

4. The authors argue that the FACS experiments in Fig S16 were done to address both

the question of replicating in the presence of HU and after removal of replication stress. The authors have shown that the mutants accumulate aberrant replication structures even early in S phase in the presence of HU and reach a different overall replication state after 4h (Fig 5B). Therefore, the recovery from HU arrest cannot be cleanly addressed with the mutants since they start from different states.

We basically agree with the reviewer that because some of the single mutants and particularly the double mutants accumulate aberrant replication intermediates by 4h in HU, it is not really possible to do an arrest and release experiment that allows a perfect comparison between the wild-type strain and the different mutants. Regardless, the results in Fig S18 clearly show that the mutants have a substantial delay in recovering from HU and retain aberrant replication intermediates even after 2h of release from HU. This is in contrast to the behavior of the wild-type strain that has fully recovered and completed replication in HU by 2h (Fig 5B) and has no delay in recovering from HU (Fig S18). Thus, we believe that this experiment, as stated in the paper, none-the-less provides useful data for readers to consider.

Reviewer #2 (Remarks to the Author):

1. In the revised manuscript, the authors have addressed many of my previous concerns. I think we will need to agree to disagree on the value of tetrad dissection compared to transformations when it comes to dealing with suppressor analyses. It does no good to argue in this forum. I think the key point is that several of the key strains have apparently been made by dissection and that several strains have also been shown to be free of second site suppressors by sequence analyses.

I am happy to agree to disagree. But I should say I do not question the value of constructing strains by crosses. Rather, I think that regardless on the strain construction method used, the strains then need to be carefully evaluated for the presence of suppressors and other unanticipated defects

2. I think the authors have not quite understood my comments with regards to an indirect, transcriptional role for Htz1. In general Htz1 only impacts induced transcription, not expression of constitutively expressed genes. Consequently, I believe (though I may be incorrect) that the data sets evaluated by the authors did not include RNA isolated from DNA damage induced cells or cells undergoing replication stress. If Htz1 is key for stress-induced gene expression, then this may be key for the phenotypes observed.

We agree that the reviewer's restated point is a possibility and we have now included a statement to this affect in the Results and Discussion on P6, L2-6 and P16, L6-9. However, we note that we consider this an unlikely explanation because there are very few replication and repair genes whose transcription is induced by replication stress and, at least under normal conditions, Htz1 defects do not appear to alter the expression of these genes. In contrast, there is direct evidence that Htz1 is directly loaded onto DNA at sites of DNA damage like DSBs. Because even after a massive transcriptional analysis, it would be impossible to exclude the possibility raised by the reviewer, I think it would be best to just simply make note of the possibility.

3. In general this is a solid study.

We appreciate the positive comment by the reviewer.

Reviewer #3 (Remarks to the Author):

1. First, I want to emphasize that I enjoyed reading this new version of the manuscript. It was much improved, including the tables. The authors have clarified how the strains were constructed and this allows each reader to make their own evaluation of the results. I do appreciate that the authors clarify how they minimize the risk for suppressors of growth defects.

We would like to thank the reviewer for their positive response to our revisions.

Reviewers' Comments:

Reviewer #1:

Remarks to the Author:

The authors have addressed all of my concerns. I am happy to support publication.

Reviewer Response

The reviewers only stated that we addressed all of their concerns so no point-by-point response is provided.